# Genetic associations with carotid intima-media thickness link to atherosclerosis with sex-specific effects in sub-Saharan Africans

Palwende Romuald Boua [1,2,3✉], Jean-Tristan Brandenburg [2], Ananyo Choudhury [2], Hermann Sorgho[1], Engelbert A. Nonterah [4,5], Godfred Agongo[4,6], Gershim Asiki [7], Lisa Micklesfield [8], Solomon Choma[9], Francesc Xavier Gómez-Olivé [10], Scott Hazelhurst [11], Halidou Tinto[1], Nigel J. Crowther[12], Christopher G. Mathew [2,3,13], Michèle Ramsay [2,3✉], AWI-Gen Study* & the H3Africa Consortium*

Atherosclerosis precedes the onset of clinical manifestations of cardiovascular diseases (CVDs). We used carotid intima-media thickness (cIMT) to investigate genetic susceptibility to atherosclerosis in 7894 unrelated adults (3963 women, 3931 men; 40 to 60 years) resident in four sub-Saharan African countries. cIMT was measured by ultrasound and genotyping was performed on the H3Africa SNP Array. Two new African-specific genome-wide significant loci for mean-max cIMT, *SIRPA* (p = 4.7E-08), and *FBXL17* (p = 2.5E-08), were identified. Sex-stratified analysis revealed associations with one male-specific locus, *SNX29* (p = 6.3E-09), and two female-specific loci, *LARP6* (p = 2.4E-09) and *PROK1* (p = 1.0E-08). We replicate previous cIMT associations with different lead SNPs in linkage disequilibrium with SNPs primarily identified in European populations. Our study find significant enrichment for genes involved in oestrogen response from female-specific signals. The genes identified show biological relevance to atherosclerosis and/or CVDs, sex-differences and transferability of signals from non-African studies.

[1] Clinical Research Unit of Nanoro, Institut de Recherche en Sciences de la Santé, Centre national de la Recherche scientifique et technologique (CNRST), Nanoro, Burkina Faso. [2] Sydney Brenner Institute for Molecular Bioscience, Faculty of Health Sciences, University of the Witwatersrand, Johannesburg, South Africa. [3] Division of Human Genetics, National Health Laboratory Service and School of Pathology, Faculty of Health Sciences, University of the Witwatersrand, Johannesburg, South Africa. [4] Navrongo Health Research Centre, Ghana Health Service, Navrongo, Ghana. [5] Julius Global Health, Julius Centre for Health Sciences and Primary Care, University Medical Centre, Utrecht University, Utrecht, The Netherlands. [6] C.K. Tedam University of Technology and Applied Sciences, Navrongo, Ghana. [7] African Population and Health Research Center, Nairobi, Kenya. [8] MRC/Wits Developmental Pathways for Health Research Unit, Faculty of Health Sciences, University of the Witwatersrand, Johannesburg, South Africa. [9] Department of Pathology and Medical Sciences, University of Limpopo, Polokwane, South Africa. [10] MRC/Wits Rural Public Health and Health Transitions Research Unit (Agincourt), School of Public Health, Faculty of Health Sciences, University of the Witwatersrand, Johannesburg, South Africa. [11] School of Electrical and Information Engineering, University of the Witwatersrand, Johannesburg, South Africa. [12] Department of Chemical Pathology, National Health Laboratory Service, Faculty of Health Sciences, University of the Witwatersrand, Johannesburg, South Africa. [13] Department of Medical and Molecular Genetics, Faculty of Life Sciences and Medicine, King's College London, London, UK. *Lists of authors and their affiliations appear at the end of the paper. ✉email: romyboua@gmail.com; michele.ramsay@wits.ac.za

Atherosclerosis is a complex multifactorial trait with an enigmatic genetic aetiology. Despite discoveries from genome-wide association studies (GWAS), little is known about the genetic contributions to atherosclerosis. Meanwhile, the worldwide epidemic of cardiovascular diseases (CVDs), including clinical manifestation of atherosclerosis, is growing and has become the leading cause of deaths worldwide[1,2]. Moreover, the health and demographic transition in sub-Saharan Africa (SSA) has shifted the major causes of death from communicable to non-communicable diseases (NCDs).

Atherosclerosis results from injury to the arterial endothelium, resulting in an inflammatory response in the vessel wall. The location and morphology of the atherosclerotic lesions predict the nature of the resulting vascular disease. Whereas family and twins studies provided evidence of high heritability of common carotid intima-media thickness (cIMT) (20-65%)[3–6], the GWAS studies reported associations that account for only 1.1% of the variance of cIMT[7]. cIMT is a widely accepted surrogate marker for the risk of generalized atherosclerosis and is a measurement used in large-scale research studies on genetic associations with future cardiovascular events[8,9].

The genetic diversity of African populations and their deep evolutionary roots represent opportunities for novel genetic discoveries. Haplotypes blocks are shorter in Africans compared to other populations (average haplotype block ~8.8 kb in Africans, ~20.7 kb in Europeans, and ~25.2 kb in Han Chinese), and therefore identification of causal variants is facilitated[10,11]. The role of ancestry in atherosclerosis risk has been established from studies in multi-ethnic settings and admixture studies[12,13]. African ancestry was reported to be associated with a higher risk of atherosclerosis compared to Europeans, Hispanics and Asians.

Since phenotypic differences between men and women are a pervasive feature of several quantitative traits, studies of sex interactions for complex human traits may shed light on the molecular mechanisms that lead to biological differences between men and women. Sex has been found to play a role in variations between gene expression and genotype across a range of human complex traits[14]. Sex differences in the transcriptomes of cells involved in the atherosclerotic process have been previously reported[15] and are supported by sex-stratified GWAS analyses[16,17]. Sex provides two different environmental contexts determined by the hormonal milieu and differential gene expression between the sexes. A recent GWA study of cIMT reported sex-specific loci from analyses of women and men from the United Kingdom BioBank (UKBB) data[18]. Despite the success of GWAS efforts, men and women have typically been analyzed together in sex-combined analyses, with sex used as a covariate in the model to account for marginal differences in traits between them. Sex-combined analyses assume homogeneity of the allelic effects in men and women, and therefore are suboptimal in the presence of heterogeneity in genetic effects by sex, i.e. sex-dimorphic effects[19,20].

Several genetic association studies of cIMT have been performed in the major world populations and provided insights into genes and tissue-specific regulatory mechanisms linking atherosclerosis both to its functional genomic origins and its clinical consequences in humans. To date, 136 SNPs from 98 independent loci have been found to be associated with cIMT (GWAS Catalog) regardless of ancestry[21]. The loci were reported for cIMT, in the presence or absence of gene-environment interactions (Gene × HIV, Gene × Smoking, Gene × Sex, Gene × Rheumatoid arthritis). The studies that are reported in the GWAS Catalog are primarily from European-ancestry populations, with small numbers of Hispanic, African-American and Chinese participants. There was only one study on sub-Saharan African populations resident in Africa[22].

The Africa Wits-INDEPTH Partnership for Genomic Studies cohort (AWI-Gen) was developed to examine genetic and environmental contributions to cardiometabolic diseases in Africans. It has over 12,000 participants from four sub-Saharan African countries, Burkina Faso, Ghana, Kenya and South Africa, and the distributions and associated risk factors for cIMT have been described[23–26]. This study aimed to investigate genetic susceptibility to atherosclerosis in sub-Saharan Africans in the AWI-Gen cohort. cIMT was used as an endophenotype, with further investigation of sex differences.

In this study, we identify five novel loci associated with cIMT in sub-Saharan African which relates to atherosclerosis' biology and showed the potential role of menopause in the pathobiology of atherosclerosis. Our study report sex-specific effects and highlights the need for more genome-wide studies in under-represented populations.

## Results

**Genetic association with cIMT**. Analyses were performed using the imputed dataset of 13.9 M SNPs in 7894 participants from the AWI-Gen study (characteristics for each study site are shown in Supplementary Data 1) and tested for association with mean-max-cIMT. Despite the population sub-structure demonstrated by principal component analysis in the study sample (Supplementary Fig. 1), our results did not show evidence of genomic inflation ($\lambda = 0.997$). The genome-wide association results for the combined dataset are illustrated in the Manhattan plot and the genomic inflation by the QQ-plot (Fig. 1a, b). In the combined dataset, we identified two new genome-wide significant loci in Signal regulatory protein alpha (*SIRPA*) on chromosome 20 (rs6045318, $p = 4.7E-08$, Info Score = 0.88) and F-box and leucine-rich repeat protein 17 (*FBXL17*) on chromosome 5 (rs552690895, $p = 2.5E-08$, Info Score = 0.97) (Table 1). These two SNPs are African-specific and the variant alleles have not been observed in European or Asian populations. The effect allele frequencies were similar in East, West and Southern Africa (respectively, 0.98, 0.96 and 0.97 for rs6045318 and 0.99, 0.98 and 0.99 for rs552690895) (Supplementary Data 2). The regional plots of the identified associated variants show the distribution of additional variants around the lead SNPs (Fig. 2A, B). Genotyped variants were distributed around the imputed lead SNPs with higher *p*-values. Suggestive association signals (p < 1-06) had lead variants located in an intergenic region on chromosome 8 (rs11781274, $p = 1.8E-07$), an intronic region in sortilin related VPS10 domain-containing receptor 1 (*SORCS1*) (rs11193156, $p = 2.1E-07$), an intronic region in ankyrin repeat and kinase domain-containing 1 (*ANKK1*) (rs11214599, $p = 5.4E-07$), an exonic region in C-terminal binding protein 2 (*CTBP2*) (rs3781409, $p = 6.6E-07$) and an intronic region in SWI/SNF related, matrix associated, actin-dependent regulator of chromatin, subfamily a, member 2 (*SMARCA2*) (rs1324201, $p = 8.6E-07$) (Supplementary Data 3).

Sex-specific analyses revealed three genome-wide significant loci (as illustrated in the Miami plots, Fig. 3): one male-specific locus led by an intronic variant in sorting nexin 29 (*SNX29*) (rs190770959, $p = 6.3E-9$) and a near significant male-specific association at mitogen-activated protein kinase kinase kinase 7 (*MAP3K7*) (rs284509, $p = 5.3E-8$), and two female-specific loci one located in an intergenic region between uveal autoantigen with coiled-coil domains and ankyrin repeats (*UACA*) and LDL receptor-related protein 6 (*LRP6*) (a downstream variant located in the promoter flanking region) (rs150840489, $p = 2.4E-09$) and a variant in a transcription factor binding site near prokineticin 1/ chymosin, pseudogene (*PROK1/CYMP*) (rs115473055, $p = 1.0E-08$) (Fig. 2C–F).

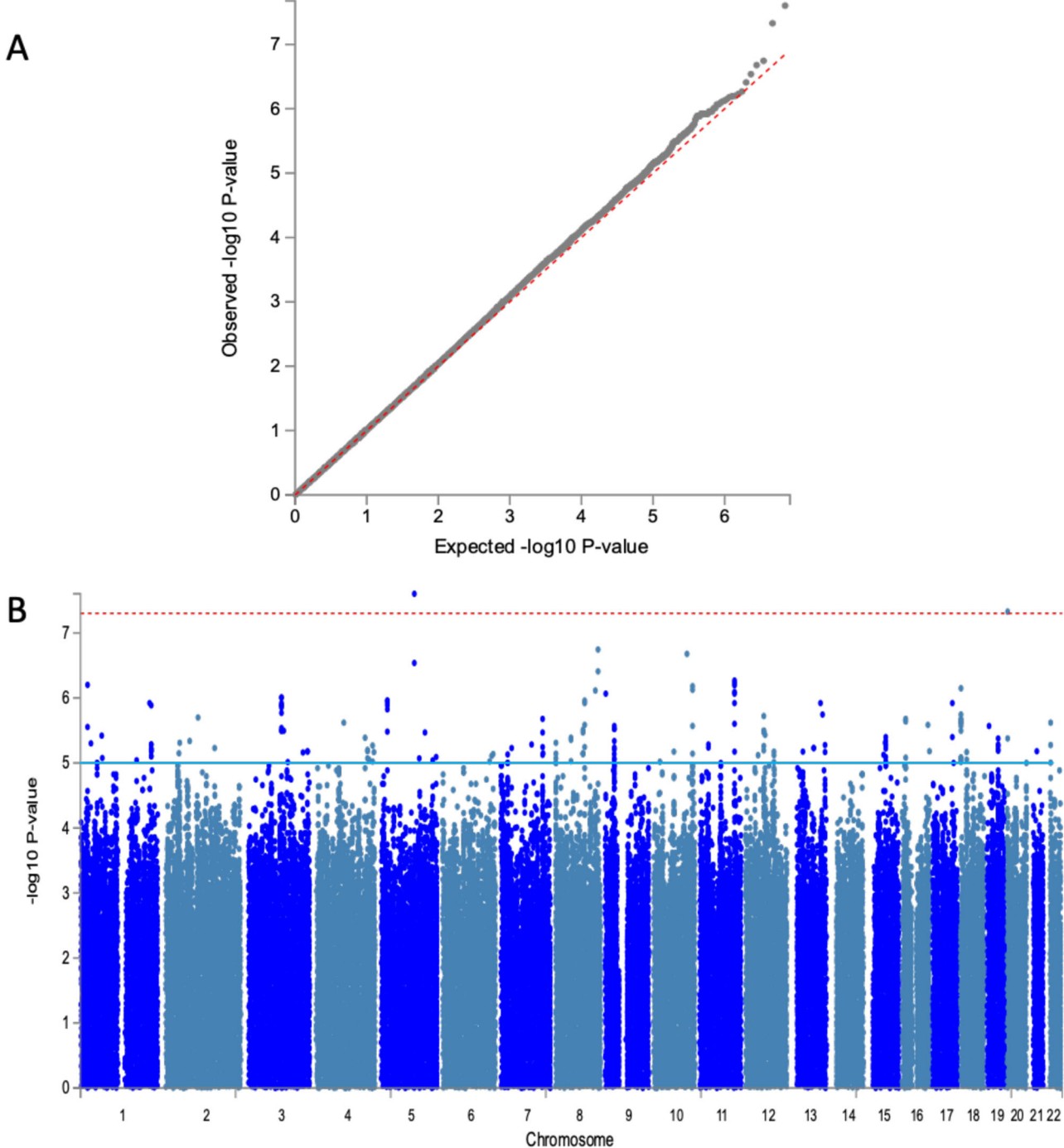

**Fig. 1 Genetic association with cIMT in sub-Saharan Africans (7894 participants). A** QQ-plot for the combined dataset GIF = 0.997. **B** Manhattan plot showing the −log10-transformed two-tailed *P*-value of each SNP from the GWAS for Mean-Max cIMT on the *Y*-axis and base-pair positions along the chromosomes on the *X*-axis. Adjustment was made for age, sex and 8 PCs. The red line indicates Bonferroni-corrected genome-wide significance ($p < 5E-08$); the blue line indicates the threshold for suggestive association ($p < 1E-05$).

Loci with suggestive sex-specific associations are shown in Supplementary Data 3).

Sex differences that were limited to men or women were assessed (Supplementary Fig. 2). We found suggestive signals in the intergenic regions of insulin-like growth factor binding protein-like 1/ family with sequence similarity 95 member C (*IGFBPL/FAM95C*) (rs12350396, $p = 3.4E-07$), *UACA/LARP6* (rs150840489, $p = 4.4E-07$), myogenic differentiation 1/ potassium voltage-gated channel subfamily C member 1 (*MYOD1/KCNC1*) (rs150481830, $p = 4.9E-07$), and in an intronic region of ciliary

rootlet coiled-coil, rootletin (*CROCC*) (rs11585710, $p = 4.8E-07$). This sex-difference are shown in Supplementary Data 4. Regional plots of significant loci are shown in Fig. 3.

Analysis of sex-dimorphism requires both a significant (or suggestive $p < 1E-04$ as applied in our study) SNP association with cIMT in at least one sex and a nominally significant association in the other sex; as well as a significant sex-difference for the SNP association (*P*-value testing for difference in sex-specific effect estimates $<1E-06$). Several scenarios can describe sexual dimorphism for SNP associations: (i) concordant effect

**Table 1 Genome-wide significantly associated SNPs ($p < 5E-08$) with Mean-Max cIMT for the combined AWI-Gen dataset ($n = 7894$).**

| GWAS | rsID | chr | pos | Non-effect allele | Effect allele | MAF | gwasP | Beta | SE | IndSigSNP | Nearest Gene | Genic position |
|---|---|---|---|---|---|---|---|---|---|---|---|---|
| Combined analysis | rs552690895 | 5 | 107570359 | A | G | 0.013 | 2.50E-08 | −0.043 | 0.008 | rs552690895 | FBXL17 | Intronic |
| | rs604518 | 20 | 1883451 | G | A | 0.024 | 4.70E-08 | −0.031 | 0.006 | rs604518 | SIRPA | Intronic |
| Female-specific | rs115473055 | 1 | 110020548 | T | C | 0.080 | 1.00E-08 | −0.026 | 0.005 | rs115473055 | CYMP | Intergenic |
| | rs150840489 | 15 | 71088277 | A | G | 0.037 | 2.40E-09 | −0.051 | 0.009 | rs150840489 | RPL29P30 | Downstream |
| Male-specific | rs190770959 | 16 | 12158574 | T | C | 0.015 | 6.30E-09 | −0.056 | 0.01 | rs190770959 | SNX29 | Intronic |
| | rs147978408 | 16 | 12171475 | C | T | 0.016 | 6.60E-09 | −0.055 | 0.01 | rs190770959 | SNX29 | Intronic |

Adjustment was made for age, sex and 8 PCs for the combined analysis and for age and 5 PCs for sex-specific analysis. Bonferroni correction for genome-wide significance was made ($p < 5E-08$).
*pos* position, *chr* chromosome, *rsID* single-nucleotide polymorphism, *MAF* minor allele frequency for the combined sample (7894), *beta* SNP effect for the combined analysis, *SE* standard error, *gwasP* p-value for the association test with BOLT-LMM, *IndSigSNP* lead SNP for the signal.

direction (CED); (ii) single sex effect (SSE); or (iii) opposite effect direction (OED)[27]. In our study, we identified all three types of sexual dimorphism: the *LARP6* locus was a case of a single sex effect (rs150840489: $p$-female = 2.4E-09, beta-female = −0.051; $p$-male = 0.17, beta-male = 0.012); the *CROCC* locus showed opposite effect direction (rs11585710: $p$-female = 4.9E-03, beta-female = 0.007; $p$-male = 2.7E-05, beta-male = −0.012), and the *FBXL17* variant showed a concordant effect direction (rs547840497, $p$-female = 0.037, beta-female = −0.022; $p$-male = 1.8E-07, beta-male = −0.062). In total 177 SNPs showed CED, 89 SNPs had OED and 3213 SNPs showed SSE.

**Replication of previous associations with cIMT.** We investigated the replication of 47 previous cIMT associations in our GWAS study (Supplementary Data 5). Although, we did not observe an exact replication, one variant (rs561732; $p = 0.0012$; beta = 0.006) was nearly significant after correction for multiple testing (Bonferroni correction for 47 SNPs; $p = 0.05$). This variant in the CBFA2/RUNX1 partner transcriptional co-repressor 3 (*CBFA2T3*) region has been reported for association with cIMT in a British ancestry population (UKBB)[18]. The association of the *CBFA2T3* locus with cIMT, based on a different variant, was first reported in a European-ancestry study[28].

In addition, four of the previously reported signals were found to be replicated in the local replication analysis which considered, for each signal, all the SNPs that were within 250 kb of the index SNP and showed a LD > 0.7 (Supplementary Data 6). This included five variants on chromosome10 (bp = 56620608–56625539) close to a variant (rs975809) previously associated with cIMT progression in a Chinese population[29]. Similarly, variants on chromosome 16 (bp = 88966667–88978850) replicated previous findings in European and British ancestry populations. For another signal on chromosome 16 detected by Strawbridge and colleagues[18], variants in close proximity (bp = 88968540–89016494) also showed local replication. Finally, we found evidence of regional replication for a variant on chromosome 8 (rs6601530), as first reported for cIMT GWAS in a European study (bp = 10584288–10722058) study[7] (Supplementary Data 6).

**Look-up for cardiovascular traits in the GWAS Catalog.** Variants with suggestive associations in our study were located in loci previously reported for traits such as plaque, coronary artery calcification (CAC), coronary artery disease (CAC), coronary heart disease (CHD), coronary aneurysm and coronary atherosclerosis. Previously reported locus for association with a carotid plaque in European populations[30] at *GEM* (rs72672639, $p = 4.0E-06$) to be suggestively associated in our female-specific subset with two SNPs (rs78571209, rs76489670, $p = 7.8E-05$) located approximatively 2200 bp from the SNP reported for plaque in Europeans. Similarly, the association with the mitochondrial ribosomal protein L37 (*MRPL37*) locus (rs11206301, $p = 8.00E-06$) for plaque in European populations was suggestively associated in our male-specific analysis for cIMT (rs13374450, $p = 3.0E-05$). The two SNPs in the *MRPL37* locus were not in LD despite their proximity (201 bp). The suggestive variant in our study rs4773141 ($p = 4.7E-05$, in the combined dataset), located in collagen type IV alpha 1 chain (*COL4A1*), was previously reported for CAD ($p = 4.0E-17$) in European populations[31].

In our combined analysis, a total of 10 suggestively associated SNPs ($p < 1E-04$) were previously described for association with CAC (another surrogate marker of atherosclerosis)[32,33] and for CAC in African patients with type 2 diabetes[34]. Fourteen SNPs reported for coronary heart disease and coronary artery disease were suggestive in our dataset (Supplementary Data 7).

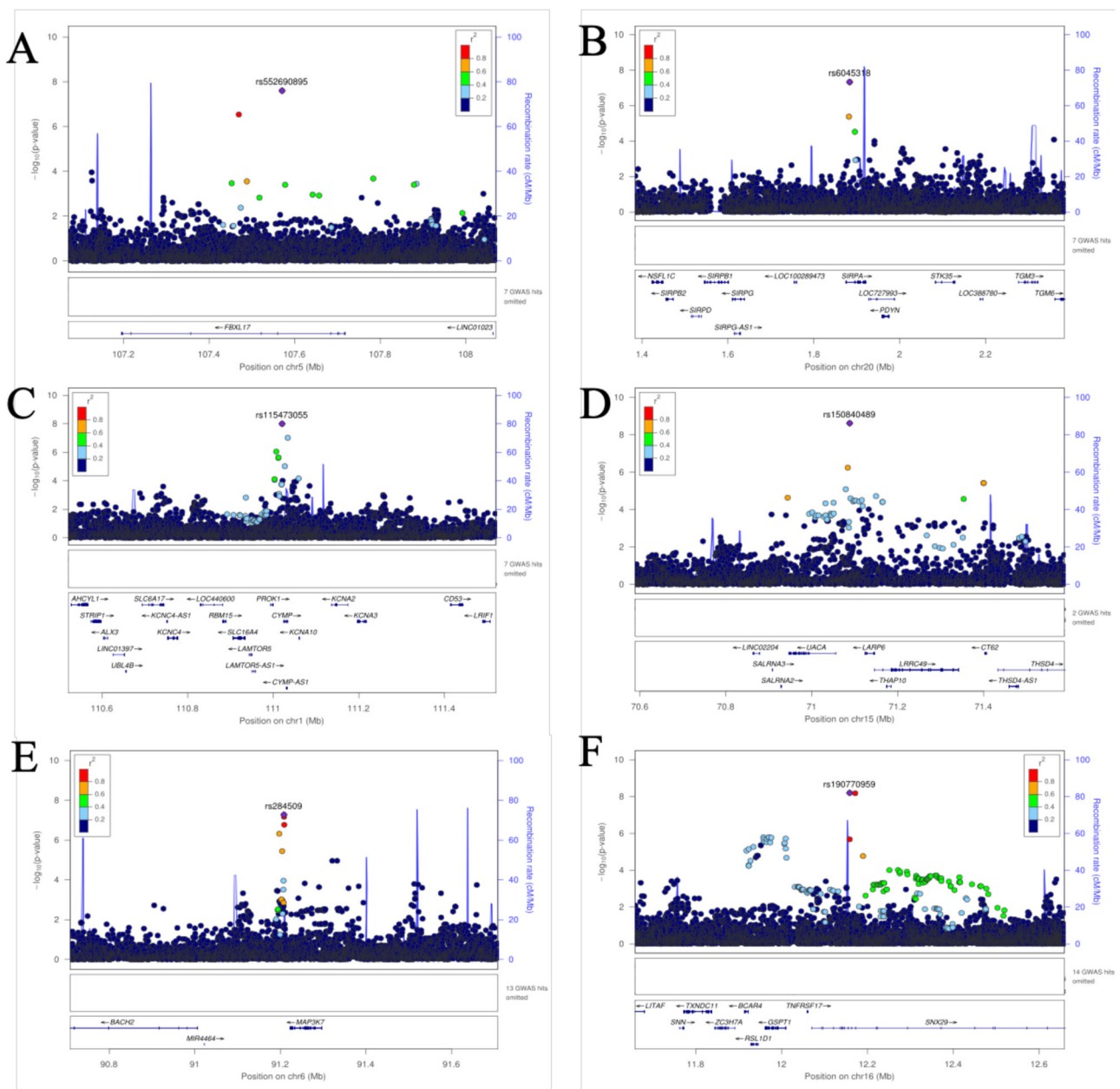

**Fig. 2 Regional association plots for selected top SNPs showing genetic associations with mean-max cIMT. A** Regional association plot of the *FBXL17* region in the combined dataset. **B** Plot of the *SIRPA* region in the combined dataset. **C** Regional association plot of the *PROK1* region in the female-specific dataset. **D** Regional association plot of the *LARP6/UACA* region female-specific dataset. **E** Regional association plot of the *MAP3K7* region in the male-specific dataset. **F** Regional association plot of the *SNX29* region in the male-specific dataset. For each locus, the plots show the −log10-transformed *p*-value of each SNP on the *y*-axis and base-pair positions along the chromosomes on the *X*-axis. Genes overlapping the locus are displayed below the plot. SNPs are coloured by their LD value r$^2$ (generated from the study population) with the lead SNP in the region shown as a purple diamond.

**Functional annotation**. Annotation of the genic positions of the 467, 515 and 581 SNPs respectively from the combined, female-specific and male-specific analyses with significant and suggestive associations ($p < 1E-05$) showed that these were mostly intronic or intergenic. Fifty SNPs displayed a CADD score above 12.37 suggesting being potentially deleterious (19 in the combined; 18 in female-specific; 13 in male-specific datasets) (Supplementary Data 8a, b, c). In the female-specific sample, the lead SNP in *CYMP* (rs115473055) had a Regulome DB score of 2a suggesting the variant was likely affecting a transcription binding site (CTCF). Positional mapping, eQTL mapping (matched cis-eQTL SNPs) and chromatin interaction mapping (on the basis of 3D DNA–DNA interactions) is reported (Supplementary Data 9a,

b, c). We found that rs78172571, in high LD with rs150840489 (the top SNP associated in our female-specific analysis), was involved in HiC type chromatin interactions in multiple tissues including aorta, in which the variant acts as an enhancer of THAP domain-containing 10 (*THAP10*) (FDR = 2.03E-17).

**Gene-based and gene-set analysis**. In a gene-based analysis (using MAGMA threshold of $p < 2.6E-06$) of the combined dataset analysis there was a significant association with *CALD1* ($p = 5.9E-07$) (Supplementary Fig. 3A) with mean-max cIMT, whereas in female-specific analysis *FLT4* ($p = 4.3E-07$) was significantly associated (Supplementary Fig. 3B). The results from gene-set analysis in the

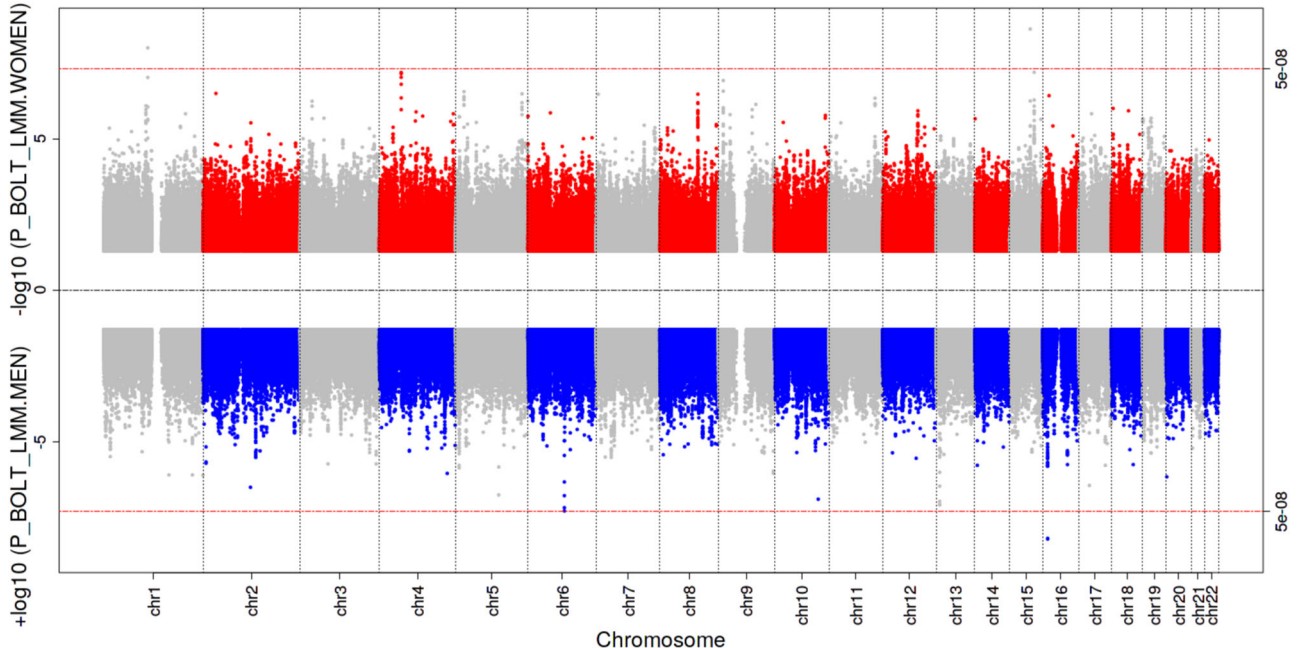

**Fig. 3 Miami plot showing female and male-specific associated *p*-values for mean-max cIMT.** The −log10-transformed two-tailed *P*-value of each SNP from the GWAS for Mean-Max cIMT on the *Y*-axis (Adjusted for age and 5 PCs for each sex) and base-pair positions along the chromosomes on the *X*-axis. On top are the results for female-specific analysis (*n* = 3963) and on the bottom the results for the male-specific analysis (*n* = 3931). The red line indicates Bonferroni-corrected genome-wide significance ($p < 5 \times 10^{-8}$).

combined dataset showed significant enrichment for "Chemical and Genetic perturbation" gene-set (adjP = 3.9E-05). The female-specific analysis revealed significant enrichment of gene-sets (Supplementary Data 10a, b, c), with among them "Hallmark gene-sets for Oestrogen response", with "Early Oestrogen response" and "Late Oestrogen response" both being significant (2.2E-6). In order to further investigate the effect of menopause in our sample, we tested the impact of menopause on female-specific significant variants (rs115473055, rs150840489) in two strata (pre-menopausal and post-menopausal women). We found that for rs115473055, the carriers of the T allele displayed higher cIMT values than those with the C allele in post-menopausal participants. There was no difference in pre-menopausal women (Fig. 4).

## Discussion

The outcome variable for our GWAS study in African populations was cIMT, often used as a proxy for the risk of developing atherosclerosis. We identified two new loci associated with cIMT in the full dataset, two new loci specific to the female-only analysis and one locus associated with the male-only analysis ($p < 5$E-08). We demonstrated that some signals observed in European populations replicate in Africans, in spite of differences in the lead associated variants, allele frequencies and effect sizes. The modest sample size of our study limited our capacity to confirm previous low-effect associated loci detected in Europeans.

Measurements of cIMT are used clinically to assess vascular pathophysiology and to reflect the risk of developing atherosclerosis. Our study identified cIMT-associated loci relevant to genes related to macrophage activity and polarization (*SIRPA*), to vascular smooth muscle cells (*MAP3K7, CALD1*), to vascular endothelial growth (*PROK1, FLT4*), to collagen synthesis and plaque stability (*LARP6*) and a pathway of blood vessel occlusion (*SNX29*). The loci for the genome-wide significant associations and their potential role in atherosclerosis are briefly described below.

*FBXL17* (lead SNP:rs552690895; $p = 2.5$E-08) was associated with cIMT in the combined dataset analysis and is linked to

cardiovascular physiology through its involvement in protein degradation where it plays a central role in cardiovascular physiology and disease: from endothelial function, the cell cycle, atherosclerosis, myocardial ischaemia, cardiac hypertrophy, inherited cardiomyopathies and heart failure. A GWAS in Lithuanian families found that variants in *FBXL17* were associated with coronary heart diseases[35]. Signal regulatory protein alpha (*SIRPA*) (lead SNP:rs6045318; $p = 4.7$E-08 in the combined analysis) has a role in the mediation of phagocytosis and polarization of macrophages which is important in the pathophysiology of atherosclerosis[36]. There is evidence that *SIRPA* is involved in discrete stages of cardiovascular cell lineage differentiation[37] and that defects in the gene (knock out) reduce atherosclerosis in mice[38]. *SIRPA* expression has been found as a signature of inflamed atherosclerotic plaque[39].

In the sex-specific analysis, the top cIMT-associated variant in men was in the *SNX29* gene (rs147978408; $p = 6.3$E-09). The sorting nexin (*SNX*) family of genes are associated with CVDs, and dysfunction of the *SNX* pathway is involved in several forms of cardiovascular disease (CVD)[40]. In a study of genes that regulate smooth muscle cell differentiation and disease risk, *SNX29* was involved in pathways for occlusion of blood vessels and atherosclerosis[41]. Ito and colleagues identified sex-dependent differentially methylated regions close to *SNX29* in mouse liver and found that methylation status was influenced by testosterone and contributed to sex-dimorphic chromatin decondensation[42]. This might explain the sex-specific effect observed in our study. In view of the previous link between *SNX29* and hypertension, we ran further GWAS analyses stratified by hypertensive status and found that the association was driven by the hypertensive group (the effect was three times higher in hypertensives compared to non-hypertensives), thereby demonstrating that the association of *SNX29* with cIMT might be mediated by the vascular remodelling caused by hypertension.

Another implicated gene, *LARP6* (La-related protein 6), is a ribonucleoprotein domain family member 6 with a role in collagen regulation by targeting mRNA encoding Type I collagen[43,44].

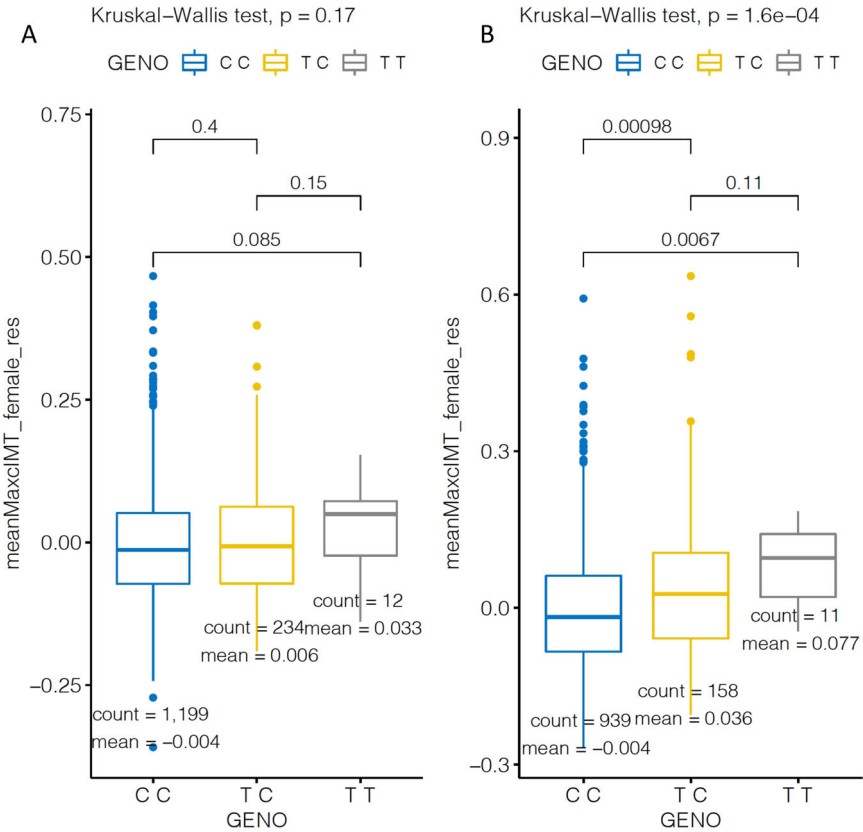

**Fig. 4 Genotype plots for genetic associations of rs115473055 stratified by menopausal status. A** Pre-menopausal women ($n = 1245$), **B** Post-menopausal women ($n = 1108$) with cIMT. Bounds of boxes represent 1st and 3rd quintiles, bars represent 95th percentiles, mean.

Collagen is a hallmark of atherosclerotic plaque stability, thus alteration of the collagen balance may lead to instability of atherosclerotic lesions, and therefore promote plaque formation and rupture[39,45]. In the Taiwanese population, the *LARP6* locus was found to be associated with coronary artery disease[46]. Myocardial gene expression in non-ischemic human heart failure found that *LARP6* was differentially expressed between men and women[47]. The female-specific effect of this locus in our study may be explained by the enhancer function of rs78172571. This SNP is in high LD with rs150840489 (the top SNP associated in our female-specific analysis), on the *THAP10* gene (FDR = 2.03E-17), known to be regulated by oestrogen[48]. Our study is the first to report prokineticin 1 (*PROK1*) for any trait in a GWAS. It was associated with cIMT in the female-specific analysis (lead SNP:rs115473055, p = 1.00E-08). *PROK1* is a specific placental angiogenic factor that plays a role in the control of normal (e.g. endometrial decidualization) and pathological placental angiogenesis[49]. The gene is known to be predominantly expressed in the steroidogenic glands, such as the ovary, testis and adrenal cortex, and is often complementary to the expression of vascular endothelial growth factor (*VEGF*), suggesting that these molecules function in a coordinated manner. The function and a particular pattern of this gene's activity might explain why we identified the locus only in our female-specific analysis. Our study revealed that rs115473055, situated in a transcription binding site and likely to affect gene expression, was subject to different allelic effects depending on the menopausal status of women. This is a novel finding, and more discussion on potential biological relevance is presented in Supplementary Note 1.

Interestingly, our study identified loci with suggestive association ($P < 1E-04$) which had opposite effects on cIMT between men and women (led by signals in the *CROCC* locus). In contrast,

the sex-stratified analysis with participants from the UKBB reported a single discordant sex effect and the remainder of the effect directions were concordant. It is possible that sex differences are amplified in African populations and are therefore noteworthy. Although statistically significant (Supplementary Fig. 2 of sex-difference test Manhattan plot), these findings need to be explored further through replication in additional studies because our power to detect the associations observed was low, substantially increasing the likelihood that the associations we observed are false positives.

When analysing sex-specific or gene-sex interactions, it is important to keep in mind that they also reflect the influences of non-genetic factors such as behaviour, as evidenced by the previously reported gene-smoking interactions[22]. Hence, environmental exposure, anatomical differences and sex hormone environment, which create systemic differences between males and females for trait expression, affect disease risk and heritability[50].

Our study identified significant enrichment of oestrogen pathway genes in our female-specific analysis. Oestrogen-dependent regulation of vascular gene expression and vascular physiology encompasses complex processes involving both nuclear and membrane-associated oestrogen signalling pathways. In recent years we have witnessed major progress in understanding how these regulatory processes contribute to the athero-protective effects exerted by oestrogens. Animal models of atherosclerosis provided compelling evidence that physiological oestrogen levels potently attenuate both early and advanced stages of atherosclerosis lesion development in females and suggested similar protective effects in males. The effect of oestrogens on atherosclerosis may be mediated by effects on metabolism (lipid, glucose), macrophage function or smooth muscle cells.

Nonetheless, hormone replacement therapy during menopause has not been shown to conclusively reduce atherosclerosis risk, suggesting that more studies are needed to fully decipher the biological mechanisms.

The two novel genome-wide significant associations with cIMT in our African study were not reported in previous studies. *SIRPA* and *FBXL17* are both biologically plausible candidates for cardiovascular diseases and there are several possible explanations for why these associations have not been identified in European and Asian populations. Firstly, the associated SNPs are monomorphic in these populations and were therefore not present in the replication datasets, secondly regional replication is affected by linkage disequilibrium and thirdly there may be unique gene-environment interactions that are at play in the African context. We found evidence of gene-set enrichment for biological processes. Our study is the first GWAS to report significant enrichment of genes in the oestrogen pathway for cIMT in our female-specific analysis. The findings from our study support the notion that genomics studies in Africa are likely to contribute to the understanding of complex traits, such as atherosclerosis.

**Strengths and limitations**. Our study is the first population-based study to investigate the genetic architecture of cIMT in sub-Saharan African populations. We used an analysis framework allowing us to identify genetic effects that point in opposite directions in men and women and to detect genetic effects that are only present or more pronounced in one stratum, a method that has been shown to have better power to identify qualitative gene-sex interactions[51]. The use of a new SNP genotyping array with a better representation of common African variants and imputation reference panels from African participants has improved the SNP coverage in ethnically diverse African populations. However, many of these additional SNPs are not present on Eurocentric GWAS arrays, and if they are monomorphic or have very low allele frequencies they will not be accurately imputed, if at all. SNP level replication would therefore be limited for such loci.

Although the threshold for genome-wide significance that would be considered appropriate for GWAS conducted in African populations is under debate[52–54], in the absence of a clear guideline we have employed the traditional genome-wide significance threshold of 5E-08. Post-hoc power calculation showed that the power to discover the six genome-wide significant variants was modest (Supplementary Data 11), suggesting the need for validation.

The absence of an ethnically matched replication cohort is a limitation in our study, and it will be important to replicate these findings in additional suitable cohorts. We identified African-specific variants in new loci and replicated previously reported loci, revealing opportunities for trans-ancestry fine-mapping.

## Methods
**Study population and phenotype assessments**. This is a cross-sectional study that investigated populations from six sub-Saharan African sites in West Africa (Burkina Faso (Nanoro) and Ghana (Navrongo)), East Africa (Kenya (Nairobi)) and South Africa (Agincourt, Dikgale and Soweto) as part of the AWI-Gen study[23,24,55–60]. The participants for this study include 10,703 black African men and women from two urban settings (Nairobi and Soweto) and four rural settings (Agincourt, Dikgale, Nanoro and Navrongo), aged 40–60 years. Participants completed a questionnaire requesting information on demography, health history and behaviour. Anthropometric measurements were taken and blood collected for genotyping (H3Africa SNP array) and phenotyping (biomarkers)[24]. Ultrasound scans were performed to assess cIMT of the right and left carotid arteries. No cIMT data was collected for female participants from Soweto because they were drawn from the Study of Women Entering and Endocrine Transition (SWEET) study for whom no cIMT data were collected, and they were therefore not included in the subsequent GWAS. This study received approval from the Human Research Ethics Committee (Medical), University of the Witwatersrand, South Africa (M121029, M1706110). All the participants provided written informed consent prior to

enrolment and participation in the study. Menopause was defined as the absence of a menstrual period for more than 12 months.

**cIMT measurement**. cIMT was measured using Dual B-mode ultrasound images of the carotid tree showing a typical double line for the arterial wall. Details of the method for measurement are provided in Ali et al.[24]. The cIMT values were QCed according to the Mannheim Consensus guidelines defining the use of cIMT in population-based studies. The mean-max cIMT was generated as the average of the maximum cIMT from the left and right carotid arteries, and this value was used for the GWAS analyses. The association of cIMT with non-genetic risk factors in our cohort have been documented[26,61].

**Genotyping and imputation**. The H3Africa genotyping array (https://chipinfo.h3abionet.org), designed as an African-common-variant-enriched GWAS array (Illumina) with ~2.3 million SNPs, was used to genotype genomic DNA using the Illumina FastTrack Sequencing Service (https://www.illumina.com/services/sequencing-services.html). The following pre-imputation QC steps were applied to the entire AWI-Gen genotype dataset. Individuals with a missing SNP calling rate greater than 0.05 were removed. SNPs with genotype missingness greater than 0.05, MAF less than 0.01 and Hardy-Weinberg equilibrium (HWE) P-value less than 0.0001 were removed. Non-autosomal and mitochondrial SNPs, and ambiguous SNPs that did not match the GRCh37 reference alleles or strands were also removed using the H3ABioNet pipeline[62]. Imputation was performed on the cleaned dataset (with 1,729,661 SNPs and 10,903 individuals) using the Sanger Imputation Server and the African Genome Resources as reference panel. We selected EAGLE2[63] for pre-phasing and the default PBWT algorithm was used for imputation. After imputation, poorly imputed SNPs with info scores less than 0.6, MAF less than 0.01, and HWE P-value less than 0.00001 were excluded. The final QC-ed imputed dataset had 13.98 M SNPs, and only participants with both good quality cIMT and genotyping data (*n* = 7894) were used for the GWAS analyses.

**Genome-wide association analysis**. Linear regression of Mean-Max cIMT was performed with covariates in R (https://www.R-project.org/). Residuals were extracted from the linear regression analyses and used for the GWAS analysis. We used as covariates age, sex and 8 principal components (PCs) computed on genetic data. In our sex-stratified analysis (3963 women, 3931 men), the covariates were age and 5 PCs. The number of PCs to include in each model was determined using stepwise regression and applied the Kaiser criteria as a stopping rule, which recommends stopping when the addition of PCs no longer increase the variance explained. We performed all association testing with the residuals in BOLT-LMM, which implements testing using a Linear Mixed Model (LMM). To run efficiently, BOLT-LMM required three components: the (imputed) genotypic data for association testing; a reference panel of LD scores per SNP, calculated using 1000 Genomes Project African samples; and genotype data used to approximate a genetic relationship matrix (GRM) (using a subset of the SNP array genotypes following LD filtering). This method is expected to account for all forms of relatedness, ancestral heterogeneity in the samples and other (potentially hidden) structure in the data. The analyses were run on the automated workflow developed by H3ABioNet (H3agwas) (http://github.com/h3abionet/h3agwas/)[62]. We screened the output for a genome-wide significance threshold (*p*-values < 5.E-08). To assess genomic inflation, we compared our observed distribution of −log10(P) values to that expected in the absence of association (Lambda) and illustrated the results in QQ plots. The same process was applied for sex-stratified analyses.

We used EasyStrata[64] to test for the joint effect calculated from sex strata results[65] and to test for the difference between the results from the two strata as a means to test for sex effects[27]. The joint and stratified frameworks were found to be the most efficient way to test for gene-environment interactions[66]. Power calculations were performed for the combined dataset, not the sex-specific analyses, using Quanto (Version 1.2.4) (http://biostats.usc.edu/Quanto.html), based on a range of previously reported effect sizes and different allele frequencies. We showed that a model that assesses the cIMT in independent individuals with an additive genetic inheritance and an allele frequency of 0.04 will be >93% powered (α = 5E-08) to detect a βG (genetic effect) of 0.0147 mm. Likewise, an allele frequency of 0.48 will have 98% power to detect even a very small genetic effect (β = 0.0067 mm) (Supplementary Fig. 4). We have performed post-hoc power calculation based on the gwas significant results, using power add (https://rpubs.com/maffleur/post-hoc-power).

**Replication from the GWAS catalog**. The GWAS Catalog database was downloaded (https://www.ebi.ac.uk/gwas/, accessed on 20 May 2021) and a subset of the data generated using the following keywords relevant to our study: coronary artery disease, carotid atherosclerosis, cIMT, coronary artery calcification and abdominal artery aneurism. In order to examine replication of previous findings, a two-stage approach was followed: 1. exact replication of lead SNPs and 2. local regional replication. For exact replication 53 variants present in the GWAS Catalog were extracted from our database and the *p*-values in our study reported. For local regional replications, only the SNP showing LD > 0.7 (in 1000 Genomes European populations) with the index (lead) SNP (in the discovery GWAS) and occurring

within 250 kb with it were tested. We searched for markers in our dataset that were from the output after Bonferroni correction was applied to determine significance.

**Functional analysis**. The FUMA online platform (http://fuma.ctglab.nl/)[67] was used to annotate, prioritize, visualize and interpret GWAS results. GWAS summary statistics ($p < 1E-05$) from out study was used as the input. FUMA provided extensive functional annotation for all SNPs in genomic areas identified by lead SNPs. From the list of gene IDs (as identified by SNP2GENE option in FUMA) FUMA annotated genes in a biological context[67]. We selected all candidate SNPs in the associated genomic region having $r^2 \geq 0.6$ (with 1000 Genome Project African references) with one of the independently significant SNPs, with a suggestive $P$-value ($p < 1E-05$) and MAF > 0.01 for annotation. Predicted functional consequences for these SNPs were obtained by matching the SNP's chromosome base-pair position, and reference and alternate alleles, to databases containing known functional annotations, including ANNOVAR[68], combined annotation-dependent depletion (CADD) scores[69], and Regulome DB (RDB)[70] scores. Additionally, eQTLs scans[71] were performed.

**Functional annotation of mapped genes**. Genes implicated by mapping of significant GWAS SNPs were further investigated using the GENE2FUNC option in FUMA[67], which provides hypergeometric tests of enrichment of the list of mapped genes in 53 GTEx tissue-specific gene expression sets[71], 7,246 MSigDB gene-sets (http://software.broadinstitute.org/gsea/msigdb), and chromatin states[72].

**MAGMA Gene-based and gene-sets analysis**. Multi-marker analysis of genomic annotation (MAGMA, v1.6) gene analysis was performed using summary statistics of our association results as input in the FUMA online platform using 1000 Genomes Project Africans LD. Gene-based analysis enabled summarizing SNP associations at the gene level and association of the set of genes to biological pathways. MAGMA employs multiple linear regression to obtain gene-based $p$-values[67,73]. The window for gene annotation was set for 25 kb and genome-wide significance was set at 0.05/number of tested genes. MAGMA gene-set analysis used a competitive testing framework, with gene-sets from MsigDB (v6.2, 10678 gene-sets (curated gene-sets: 4761, GO terms: 5917))[74]. MAGMA analysis was implemented within FUMA.

**Reporting summary**. Further information on research design is available in the Nature Research Reporting Summary linked to this article.

## Data availability

The processed data generated in this study are provided in the Supplementary Information. The AWI-Gen data used in this study are available to interested researchers through EGA, subject to controlled access review by the Data and Biospecimen Access Committee of the H3Africa Consortium. AWI-Gen (EGA00001002482) phenotype dataset is available at study number EGAD00001006425. AWI-Gen genotype dataset accession number: EGAD00010001996. GWAS Catalog (https://www.ebi.ac.uk/gwas/). Summary statistics reported in the paper are accessible on GWAS Catalog at the accession numbers: GCST90092502, GCST90092503, GCST90092504, GCST90092505.

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

## Acknowledgements

This study would not have been possible without the generosity of the participants who spent many hours responding to questionnaires, being measured and having samples taken. We wish to acknowledge the sterling contributions of our field workers, phlebotomists, laboratory scientists, administrators, data personnel, and all other staff who contributed to the data and sample collections, processing, storage and shipping. Investigators responsible for the conception and design of the AWI-Gen study include the following: MR (PI, Wits), Osman Sankoh (co-PI, INDEPTH), Stephen Tollman and Kathleen Kahn (Agincourt PI), Marianne Alberts (Dikgale PI), Catherine Kyobutungi (Nairobi PI), HT (Nanoro PI), Abraham Oduro (NavrongoPI), Shane Norris (Soweto PI) and SH, Nigel Crowther, Himla Soodyall and Zane Lombard (Wits). We would like to acknowledge each of the following investigators for their significant contributions to this research, mentioned according to affiliation: Wits AWI-Gen Collaborative Centre: Stuart Ali, AC, SH, Freedom Mukomana, Cassandra Soo; Soweto (DPHRU): Nomses Baloyi, Yusuf Guman. This study was funded by the National Institutes of Health (NIH) through the H3Africa AWI-Gen project (NIH grant number U54HG006938) and the Wits Non-Communicable Disease Research Leadership Programme (NIH Fogarty International Centre grant number D43TW008330). AWI-Gen is supported by the National Human Genome Research Institute (NHGRI), Eunice Kennedy Shriver National Institute of Child Health & Human Development (NICHD), Office of the Director (OD) at the National Institutes of Health. P.R.B. was funded by the National Research Foundation/The World Academy of Sciences "African Renaissance Doctoral Fellowship" (Grant no. 100004).

## Author contributions

P.R.B., H.S., H.T., A.C., C.M. and M.R. designed the study. P.R.B. and J.-T.B. performed the analysis. P.R.B. wrote the manuscript. P.R.B., J.T.B., H.S., H.T., A.C., C.M., M.R., G.Ag., G.As., E.A.N., L.M., S.C., F.X.G., S.H. and N.J.C. critically reviewed and approved the manuscript.

## Competing interests

The authors declare no competing interests.

## Additional information

## AWI-Gen Study

Palwende Romuald Boua [1,2,3 ✉], Jean-Tristan Brandenburg [2], Ananyo Choudhury [2], Hermann Sorgho[1], Engelbert A. Nonterah [4,5], Godfred Agongo[4,6], Gershim Asiki [7], Lisa Micklesfield [8], Solomon Choma[9], Francesc Xavier Gómez-Olivé [10], Scott Hazelhurst [11], Halidou Tinto[1], Nigel J. Crowther[12] & Michèle Ramsay [2,3 ✉]

## the H3Africa Consortium

Palwende Romuald Boua [1,2,3 ✉], Jean-Tristan Brandenburg [2], Ananyo Choudhury [2], Hermann Sorgho[1], Engelbert A. Nonterah [4,5], Godfred Agongo[4,6], Gershim Asiki [7], Lisa Micklesfield [8], Solomon Choma[9], Francesc Xavier Gómez-Olivé [10], Scott Hazelhurst [11], Halidou Tinto[1], Nigel J. Crowther[12] & Michèle Ramsay [2,3 ✉]

