## [Peer Review File · Nature Communications]

Genetic associations with carotid intima-media thickness link to atherosclerosis with sex-specific effects in sub-Saharan AfricansReviewers' Comments:

Reviewer #1:

Remarks to the Author:

Boua et al conducted a genome-wide association study of ultrasound-measures of carotid intima-media thickness (cIMT) in 7894 adults from four sub-Saharan African countries. Found 5 loci passed the genome-wide threshold of $p < 5 \times 10^{-8}$ (i.e., 2 low-frequency variants from the sex-combined analysis (rs552690895: MAF=1.3%, FBXL17 and rs6045318: MAF=2.4%, SIRPA), 2 African-specific common variants from the female-stratified analysis (rs115473055: MAF 8%, CYMP, rs150840489: MAF 3.7%, intergenic between UACA and LARP6) and 1 African-specific low-frequency variant from the male-specific analysis (rs190770959 and rs147978408: MAF 1.5%, SNX29). As the first GWAS on cIMT from sub-Saharan Africa identifying novel signals the study bears potential in understanding the pathobiology of atherosclerosis, and the size of the cohort is an important strength. As detailed below, my major concerns are: 1) The methodology for regional/local replication, 2) the interpretability of direct comparisons between GWAS signals of other traits with cIMT, 3) the extent to which population structure and absence of replication samples is addressed merits thoughtful consideration (e.g. meta-analysis of four geographic-specific GWAS), 4) over-emphasis on suggestive findings including the entire finding on sex-differences (sexual dimorphism) takes a lot of space in the text and illustrations. More details below.

1. In both instances, the cIMT risk-allele is the major allele (with frequency $>97\%$ in the study population). In several African populations from the 1000G data, the risk allele frequencies of the two loci from the sex-combined analysis and the locus from the male-stratified analysis (rs552690895, rs6045318, rs190770959/rs147978408) are $\sim 100\%$, suggesting the variants are rare. Hence, I urge the authors to verify the MAF of the SNPs in the study samples: Would the authors include separate MAF for samples from the 4 geographic regions? If these are imputed SNPs, what were their quality metrics?
2. For local replication, the authors included SNPs within 25kb and declared significance for those with $p < 10^{-4}$. They also looked up all SNPs with $p < 10^{-4}$ and within 250kb. A problem with these approaches is that the SNPs within the defined boundary may not be in LD with the index SNP in the discovery European population; hence a significant finding may not be a correct proxy for the GWAS locus. As an example, the CBFA2T3 SNP rs9934287 that was reported as a suggestive replication of the rs844396 SNP signal reported by Franceschini et al. However, rs9934287 is also reported to be monomorphic in Europeans. rs9934287 cannot proxy (is not in LD) with the European signal (rs9934287) associated with cIMT, hence cannot be considered a replication. There are other similar instances in the paper where a SNP not in LD with the original signal was considered to be a suggestive regional replication. Therefore, all findings of the replication analysis and the conclusion "...We ...replicated previously reported loci, revealing opportunities for trans-ancestry fine-mapping." do not appear to be correct. Second, the basis for using 10^{-4} is not clear; a locus-specific criteria should be adapted taking into account autocorrelations between SNPs to determine the multiple-testing correction threshold p-value.
3. The authors acknowledged existence of population stratification based on the PCA plots as well as absence of comparable datasets for testing replication. To address these limitations (and the possible allele freq differences of the top signals as seen in existing databases, see comments #1), I encourage the authors to rather run separate GWAS by country followed by random-effect meta-analysis approaches that account for allelic heterogeneity (rather than the pooled analysis reported).
4. The section on "Replication" is based on comparison of SNPs associated with non-cIMT traits with the current cIMT summary stat. However, such comparison is difficult to interpret. One can test, and would be more informing, to evaluate whether the cIMT and related traits have shared underlying pathogenesis (shared causal signals) using multi-trait colocalization approaches such as moloc or HyPrColoc (Nature Communications 12,764 (2021)).
5. Sexual dimorphism was tested genome-wide. However, based on Table 4 and Supplementary Figure 2, none of the loci achieved the significance threshold expected and the criteria set (i.e., genome-wide significance in at least once sex) to claim differences. Despite this, line 154 & 155 say "In total 177 SNPs showed CED, 89 SNPs had OED and 3213 SNPs showed SSE." This claim does not

seem consistent with the criterion set. The methods section needs clarity about the analysis approach on sex-difference. Were SNPs that were significant in the sex-combined / sex-based meta-analyzed summary statistics selected for the sex-diff analysis after some screening (e.g. $p < \text{some cutoff}$)? Was multiple-testing applied to declare significance?

6. Several suggestive signals have been extensively described in the manuscript text, Tables and Figures (results, discussion, supplement; example Tables 1, 2, and 3 have several suggestive loc).

7. Line 345-348: Our sex-specific analyses revealed loci that support the hypothesis that sex differences may be due to sex-specific epigenetic modification, independent of sex hormone levels. It is not clear which finding is the basis for this claim? In fact the gene-based analysis showed hormonal hall mark gene-set enrichment in females (e.g, the sex hormone estrogen).

8. Power calculations were performed with Quanto (Version 1.2.4). (<http://biostats.usc.edu/Quanto.html>). I assume this is referring to the sex-specific analysis. However, I have not seen power calculation results for either the combined or sex-stratified analyses.

9. Line 439 says, "The number of PCs to include in each model were determined using a stepwise regression". Please clarify the "stopping rule" to determine the # of PCs for adjustment as these vary in sex-combined and sex-stratified analyses.

Reviewer #2:

Remarks to the Author:

This study examines the association of genetic variants on cIMT in 7894 unrelated adults (3963 women, 3931 men) aged 40 to 60 years resident in six sub-Saharan African countries using the H3Africa genotyping array.

In a GWAS two African specific genome-wide significant loci were identified in men and two genome-wide significant loci in women and two genome-wide significant loci in a combined dataset. All of these are novel, but there is a lack of an ethnically matched replication cohort.

The Gene based analysis identified two additional loci associated with cIMT.

The paper is generally well written although not easy to go through in particular in section called "Replication with GWAS catalog", where a more concise tabulation would help.

1. The sample is comprised of six sub-Saharan African sites in West Africa, East Africa, and South Africa, all together 7894 individuals aged between 40 and 60 years. It would be of interest to see this tabulated and broken up into numbers of men and women with age range from these different areas.

2. There is different genetic substructure between the different populations that comprise the sample as shown in supplementary figure 1. The GWAS analysis appropriately adjusts for age, sex and 8PCs. It would be of importance to see the associations of the identified SNPs with cIMT for each site in, for example forest plots to see if the associations were equal at all sites/countries. Despite a relatively small study sample this would be of interest.

3. The authors need to be more careful and accurate in the description of replication from the GWAS catalog. Formal replication attempt should be limited to cIMT. The reader would benefit from concise tabulation of the results from replication attempt of cIMT SNPs from other GWA studies.

4. Examination of association of other GWAS hits for other traits, albeit related traits, they are not equivalent to cIMT and may have to a large extent, other underlying genetic components as evident from a number of GWA studies. Although the examination of association of genome wide significant SNPs for those traits with the GWAs results from the present study for cIMT warrants full attention and is of interest it is in no way a formal replication. This needs to be corrected as it is misleading and suggest mechanism that is far from being proven except by the similarity of phenotypes reflecting

atherosclerosis and its consequences. I suggest renaming the subheading "Replication with GWAS catalog" to reflect more accurately what is done.

5. Results from the MAGMA analysis described revealed in the female-specific analysis significant enrichment of gene-sets including "Hallmark gene-sets for Oestrogen response", with "Early Oestrogen response" and "Late Oestrogen response" both being significant". Taking this a bit further it is known that atherosclerosis progresses rapidly in women after menopause. There should be considerable number of women who have entered menopause in this sample aged between 40 and 60 years. There was no direct information on menopause collected in this study according to the study data dictionary. However, anticipating that the menopause is occurring at a very similar age in African women as in women f ex in the US (including African American women) it would be possible to examine the effect of those SNPs on mean max cIMT stratifying on average age of menopause.

6. minor comment is that all supplemental Tables need descriptive heading and directive notes.

Reviewer #3:

Remarks to the Author:

Boua et al. report the largest GWAS to date of carotid intima-media thickness (cIMT) in populations of African origins through the Africa Wits-INDEPTH Partnership for Genomic Studies cohort (AWI-Gen) which includes >12k participants from 4 sub-Saharan African countries. The current report includes a subset of nearly 8,000 participants with both genotype data using the H3Africa genotyping array followed by imputation onto the African Genome Resources panel (leading to ~14 million SNPs imputed after QC) and Mean Max cIMT measures (average of the maximum cIMT from the left and right). The investigators highlight the identification of 2 new genome-wide significant loci in the main analyses and 4 more 'sex-specific' loci.

Major strengths of the study include the populations under study given how overwhelmingly under-represented African populations have been in genomic studies to date, the availability of a high-density of genotyped markers to match the genetic diversity of the population (as well as to maximize the quality of the imputation when combined with the an African Genome Resources panels), the sample size in the context of the effort needed to conduct the study, the use of BOLT LMM to perform analyses and guard against population stratification, and the compelling biology of the mapped genes identified by the GWAS.

That being said, there is room for improvement of the report as suggested below:

1) Title/abstract/intro: The link between cIMT measures alone and the biology of atherosclerosis is overstated. The potential contribution of one-time cIMT measure to risk of MI and stroke is modest with hazard ratios in the ~1.1 range (PMID: 22910757) and repeated cIMT measures have not been shown to be predictive of clinical events (PMID: 22541275, 29649236). These findings should be mentioned and the utility of cIMT in clinical practice should be briefly reviewed.

2) Intro: "To date, 76 SNPs have been found to be robustly associated with cIMT" – it would be helpful to summarize these findings better up front for the reader. How many loci do these SNPs represent? How many of these are also established loci for atherosclerosis (e.g. for CAD or ischemic stroke or PAD). Which racial/ethnic groups were these discoveries made in?

3) Intro/rest of manuscript: I do not find the argument for sex-stratified analyses in the intro compelling. All but one of the references (Rawlik et al. 2016, Franconi et al. 2017, and Dong et al. 2015, and Lin et al. 2015) are low impact reports that do not provide convincing evidence that the biology of cIMT is different among males vs. females. The most interesting reference is the Rawlik paper but that paper does not include cIMT as one of the phenotypes. Furthermore, the genetic correlations between males and females of a majority of the traits included in that study were still very high (>0.9) even if they were found to be statistically different from 1. This suggests differences

between the sexes are quite subtle. The arguments for a sex-stratified analyses thus need to be tempered.

4) Methods: The authors refer to a design/methods paper for the cIMT protocol used (Ali et al.). It would be helpful if the authors could mention (or reference) the presence of expected cross-sectional associations between atherosclerosis risk factors (e.g. age, sex, lipid levels, blood pressure) and the cIMT measures in the cohorts. This would provide some reassurance of the data quality of the cIMT is high.

5) Methods/results: In relation to #3, the risk of false positive genetic associations increases as the sample size decreases. The investigators mention power calculations with QUANTO for gene-environment, but they do not actually present these calculations. Post-hoc power calculations for the main effect positive associations observed in the combined and the sex-specific analyses need to be presented given the likely low variance explained by each variant. I calculated power for identifying a variant explaining 0.5% of the variance of a quantitative trait using the Purcell's genetic power calculator (<http://zzz.bwh.harvard.edu/gpc/>) and found very low power to detect such associations with the sample sizes available here. Low power increases the risk of false positives (see Wacholder et al. PMID: 15026468). Thus, power calculations need to be presented so the reader can assess probability of a false positive given the proportion of variance of cIMT explained by each SNP.

6) Results: Arguing for the presence of genetic effects in one sex that are in the opposite direction in the other sex (e.g CROCC) for cIMT is challenging given no risk factors or genetic findings to date exist that support such findings for any manifestation of atherosclerosis. The possibility of these findings being false positives should be included in the presentation of these results.

7) Methods: Have the authors considered a different p-value for genome wide significance in this African population given the overall higher number of variants genotyped/imputed combined with the lower LD? See for example arguments/data in PMID 26733288/27305981. Several of the findings are just barely genome wide significant using the traditional cut of 5×10^{-8} .

8) Results: The first part of the Results section tends to regurgitate findings in the tables by listing all data already in Tables 1-4. First, I suggest that this is minimized. Second, I believe it would be easier for the reader if only the significant findings are presented in one table and all the suggestive findings are sent to the supplement. The sex-difference findings are not that compelling (Table 4) given low power to detect differences between the groups.

9) Results/Discussion: The term "replication" is used too liberally throughout the manuscript including in the first paragraph of the discussion section when comparing results of this cIMT GWAS to those of "related" phenotypes in the GWAS catalog. Replication should only be used for findings from other studies using the exact same phenotype of cIMT. All other findings are simply evidence of pleiotropy which might be vertical or might be horizontal.

10) The authors should consider a genetic correlation study between their findings and published findings in Europeans for cIMT (and related phenotypes) using such tools as POPCORN (PMID: 27321947). Such findings would be much more convincing of "replication" (or more specifically genetic overlap of biology). Similar genetic correlation analyses could be considered for males vs females within this study. Such analyses would have more power than single SNP analyses.

11) The Discussion is too long with a large section of it repeating the information provided in Figure 4 with even more detail for each locus in the extended discussion. It would be better if the authors could summarize the key themes of function implicated by the genes so as to summarize/complement the info provided in Figure 4/supplement.

12) The "collider bias" argument is unclear to me. What exactly is the collider and what evidence is there that the collider leads to distorted associations in males vs females when one adjusts for sex?

We thank the reviewers for these pertinent comments and have addressed the concerns point-by-point below.

REVIEWER #1 REPORT:

Boua et al conducted a genome-wide association study of ultrasound-measures of carotid intima-media thickness (cIMT) in 7894 adults from four sub-Saharan African countries. The study identified 5 loci passing the genome-wide threshold of $p < 5 \times 10^{-8}$ (i.e., 2 low-frequency variants from sex-combined analysis (rs552690895: MAF=1.3%, FBXL17 and rs6045318: MAF=2.4%, SIRPA), 2 African-specific common variants from the female-stratified analysis (rs115473055: MAF 8%, CYMP, rs150840489: MAF 3.7%, intergenic between UACA and LARP6) and 1 African-specific low-frequency variant from the male-specific analysis (rs190770959 and rs147978408: MAF 1.5%, SNX29). As the first GWAS on cIMT from sub-Saharan Africa identifying novel cIMT-associated loci, the study bears potential in understanding the pathobiology of atherosclerosis. For a GWAS from sub-Saharan Africa, the size of the cohort is an important strength. As detailed below, my major concerns are: 1) the methodology for regional/local replication, 2) the interpretability of direct comparisons between GWAS signals of other traits with cIMT, 3) the extent to which population structure and absence of replication are addressed merit thoughtful re-consideration, 4) suggestive findings including the entire finding on sex-differences (sexual dimorphism) are over-emphasized, taking a lot of space in the text and illustrations. More details below.

1. In several African populations from the 1000G data, the risk allele frequencies of the two loci from the sex-combined analysis and the one locus from the male-stratified analysis (rs552690895, rs6045318, rs190770959/rs147978408) are ~100%. This suggests that the variants are likely to be rare in many SSA populations. I urge the authors to verify the MAF of the SNPs in the study samples: Would the authors include separate MAF for samples from the 4 geographic regions? If these are imputed SNPs, what were their quality metrics?

We understand the concerns of the reviewer regarding the fact that the variants are likely to be rare. To clarify this point we have added the allele frequencies for the East, West and South regions of Africa for those variants together with their quality metrics for the imputed SNPs (see text). The text in the results section has been amended to include this information: "In the combined dataset, we identified two new genome-wide significant loci in SIRPA on chromosome 20 (rs6045318, $p = 4.7E-08$, Info Score = 0.88) and FBXL17 on chromosome 5 (rs552690895, $p = 2.5E-08$, Info Score = 0.97) (Table 1). These two SNPs are African-specific and the variant alleles have not been observed in European or Asian populations. The effect allele frequencies were similar in East, West and Southern Africa (respectively 0.98, 0.96 and 0.97 for rs6045318 and 0.99, 0.98 and 0.99 for rs552690895). The regional plots of the identified associated variants show the distributions of additional variants around the lead SNPs (Figure 2A, 2B). Genotyped variants were distributed around the imputed lead SNPs with higher p-values."Page 5

2. For local replication, the authors included SNPs within 25kb and declared significance for those with $p < 10^{-4}$. They also looked up all SNPs with $p < 10^{-4}$ and

within 250kb. A problem with these approaches is that the SNPs within the defined boundary may not be in LD with the index SNP in the discovery European population; hence a significant in the current cohort may not be a true proxy for the European GWAS locus. As an example, the CBFA2T3 SNP rs9934287 was reported as a suggestive replication of the rs844396 SNP signal reported by Franceschini et al. However, rs9934287 has also been reported to be monomorphic in Europeans. Therefore, rs9934287 cannot proxy (is not in LD) with the European signal (rs844396) associated with cIMT, hence cannot be considered a replication. There are other similar instances in the paper where a SNP not in LD with the original signal was considered to be a suggestive regional replication. Therefore, all findings of the replication analysis and the conclusion "...We ...replicated previously reported loci, revealing opportunities for trans-ancestry fine-mapping" may not be truly replication. Second, the basis for using 10⁻⁴ is not clear; a locus-specific criteria should be adapted taking into account autocorrelations between SNPs to determine the multiple-testing correction threshold p-value.

We thank the reviewer for this important insight. On average, LD blocks tend to be larger in European populations, compared to African populations, but it is important to understand the boundaries of LD when considering replication. In terms of replicating a region for association, the associated SNPs in this region are likely to be different and some may be monomorphic in the other population. We agree here that LD would be an important consideration when doing replication analysis. To be more accurate in terms of replication, we have therefore revised our approach for local regional replication and taken LD into consideration. We have amended the replication sub-section of the method: " In order to examine replication of previous findings, a two-stage approach was followed: 1. exact replication of lead SNPs and 2. local regional replication. For exact replication 53 variants present in the GWAS Catalog were extracted from our database and the p-values in our study reported. For local regional replication, only the SNP showing LD>0.7 (in 1000 Genomes European populations) with the index (lead) SNP (in the discovery GWAS) and occurring within 250 kb were tested. We searched for markers in our dataset that were from the output after FDR and Bonferroni correction was applied to determine significance." . Page 13

3. The section on "Replication" is based on comparison of SNPs associated with non-cIMT traits with the current cIMT summary statistics. However, such comparison is difficult to interpret. One can test, and would be more informing, whether the cIMT and related traits have shared underlying pathogenesis (shared causal signals) using multi-trait colocalization approaches such as moloc or HyPrColoc (Nature Communications 12,764 (2021)).

We agree with the reviewer and accordingly have only included the replication of cIMT signals in the current version of the draft. The whole replication section in the results has been rewritten. Page 6

4. The authors acknowledged existence of population stratification based on the PCA plots as well as absence of comparable datasets for testing replication. To address these limitations (and the possible allele frequency differences of the top signals as

seen in existing databases, see comments #1), I encourage the authors to rather run separate GWAS by country followed by random-effect meta-analysis approaches that account for allelic heterogeneity (rather than the pooled analysis reported).

Thank you for highlighting these important points and suggesting alternate approaches to the analysis models. We agree with the reviewer that allele frequencies can introduce heterogeneity and need to be carefully considered. To assess the impact of this GWAS process, we also performed meta-analysis using the summary statistics per region of associations from the South, East and West African subsets. The comparison of the results from the two approaches show that the associated P-values and effect sizes detected using the output from the two approaches were comparable and highly correlated. As recent studies on multi-ethnic GWAS, such as the study by the PAGE Consortium have shown that a combined analysis or 'mega-analysis' could bring more power compared to a meta-analysis based approach we have presented the results based on the former approach, which is why we decided to analyse our full dataset. Results per region for the top variants in our combined analysis has been added in Supplementary Table 2.

5. Sexual dimorphism was tested genome-wide. However, based on Table 4 and Supplementary Figure 2, none of the loci achieved the significance threshold expected and the criteria set (i.e., genome-wide significance in at least once sex) to claim differences. Despite this, line 154 & 155 say “In total 177 SNPs showed CED, 89 SNPs had OED and 3213 SNPs showed SSE.” This claim does not seem to be consistent with the criterion set. The methods section needs clarity about the analysis approach on sex-difference. In addition, the methods section as presented here is not clear. Was some sort of pre-screening (e.g. $p < \text{some cutoff}$) applied on SNPs that were significant in the sex-combined / sex-based meta-analyzed summary statistics, before running sex-diff analysis? Was multiple-testing applied to declare significance?

We thank the reviewer for pointing out that we have not been clear in accurately describing the methods of the sex-specific analyses and have rewritten this section. We decided to also report suggestive results for sex-differences and have clarified which cut offs were used.

The female-specific signal on chromosome 15 showed a p-value for sex-difference of $4.38E-07$. No pre-screening was performed as suggested by Winkler and collaborators in order to maximise identification of qualitative gene-sex interactions (OED). We used this analysis framework to identify genetic effects that point in opposite directions in men and women and to detect genetic effects that are only (or more pronounced) in one stratum, a method that has been shown to have better power to identify qualitative gene-sex interactions (Winkler *et al.*, 2017). This is described on Page 11

6. Several suggestive signals have been extensively described in the manuscript text, Tables and Figures (results, discussion, supplement; example Tables 1, 2, and 3 have several suggestive loc). I suggest taking these out.

We agree with the reviewer that they need less prominence in the main text of the paper and the extensive description of the signal at *MAP3K7* was moved to the Supplementary Notes.

7. Line 345-348 states, “Our sex-specific analyses revealed loci that support the hypothesis that sex differences may be due to sex-specific epigenetic modification, independent of sex hormone levels.” It is not clear which finding is the basis for this claim? In fact, the gene-based analysis showed hormonal hall mark gene-set enrichment in females (e.g, the sex hormone estrogen).

To clarify, this statement was based on the previous discussion on the biological relevance of *LARP6* signal. We have rephrased this as follows: “The female-specific effect of this locus in our study may be explained by the enhancer function of rs78172571 which is in high LD with rs150840489 (the top SNP associated in our female-specific analysis) in the *THAP10* gene (FDR = 2.03E-17), known to be regulated by oestrogen.” Page 10

8. It has been stated that “Power calculations were performed with Quanto (Version 1.2.4). <http://biostats.usc.edu/Quanto.html>.” I assume this is referring to the sex-specific analysis. However, I have not seen power calculation results for either the combined or sex-stratified analyses.

Thank you for pointing out that we were not specific on this point. The power calculation was performed for the combined analysis, not for the sex-specific analysis. We have added the following text in the method section for more clarity: “A model that assesses the cIMT in independent individuals with an additive genetic inheritance and an allele frequency of 0.04 will be > 93% powered ($\alpha = 5 \times 10^{-8}$) to detect a β_G (genetic effect) of 0.0147 mm. Likewise, an allele frequency of 0.48 will have 98% power to detect even a very small genetic effect ($\beta = 0.0067$ mm).” Page 13

9. Line 439 says, “The number of PCs to include in each model were determined using a stepwise regression”. Please clarify the “stopping rule” to determine the # of PCs for adjustment as these vary in sex-combined and sex-stratified analyses.

We used the Kaiser criteria as a stopping rule, which recommends stopping when the addition of PCs is no longer increasing the variance explained. The text has been amended as follows: “The number of PCs to include in each model was determined by using a stepwise regression and we applied the Kaiser criteria as a stopping rule, which recommends stopping when addition of PCs no longer increase the variance explained.” Page 13

Reviewer #2 (Remarks to the Author):

This study examines the association of genetic variants on cIMT in 7894 unrelated adults (3963 women, 3931 men) aged 40 to 60 years resident in six sub-Saharan African countries using the H3Africa genotyping array.

In a GWAS two African specific genome-wide significant loci were identified in men and two

genome-wide significant loci in women and two genome-wide significant loci in a combined dataset. All of these are novel, but there is a lack of an ethnically matched replication cohort.

The Gene based analysis identified two additional loci associated with cIMT.

The paper is generally well written although not easy to go through in particular in section called “Replication with GWAS catalog”, where a more concise tabulation would help.

1. The sample is comprised of six sub-Saharan African sites in West Africa, East Africa, and South Africa, all together 7894 individuals aged between 40 and 60 years. It would be of interest to see this tabulated and broken up into numbers of men and women with age range from these different areas.

We thank the reviewer for the comments. A Supplementary Table 1 has been added to describe the samples and characteristics from each of the study sites.

2. There is different genetic substructure between the different populations that comprise the sample as shown in supplementary figure 1. The GWAS analysis appropriately adjusts for age, sex and 8PCs. It would be of importance to see the associations of the identified SNPs with cIMT for each site in, for example forest plots to see if the associations were equal at all sites/countries. Despite a relatively small study sample this would be of interest.

Thank you for the comment which also links in with a comment from Reviewer 1. A Supplementary Table 2 has been added to present results for the GWAS outcome of the top two variants for the three African regions.

3. The authors need to be more careful and accurate in the description of replication from the GWAS catalog. Formal replication attempt should be limited to cIMT. The reader would benefit from concise tabulation of the results from replication attempt of cIMT SNPs from other GWA studies.

This comment was also raised by Reviewer 1 and we agree that our previous approach was somewhat confusing. The replication section has been rewritten for the methods and now only includes cIMT GWAS for comparison. This has improved clarity throughout the methods, results and discussion sections. There is now a sub-section entitled “Replication of previous associations with cIMT” at Page 6.

4. Examination of association of other GWAS hits for other traits, albeit related traits, they are not equivalent to cIMT and may have to a large extent, other underlying genetic components as evident from a number of GWA studies. Although the examination of association of genome wide significant SNPs for those traits with the GWAS results from the present study for cIMT warrants full attention and is of interest it is in no way a formal replication. This needs to be corrected as it is misleading and suggest mechanism that is far from being proven except by the similarity of phenotypes reflecting atherosclerosis and its consequences. I suggest renaming the subheading “Replication with GWAS catalog” to reflect more accurately what is done.

We agree and thank the reviewer for this suggestion. Accordingly, the replication section has been rewritten and a new sub-section header has been included to improve clarity: “Look-up for cardiovascular traits in the GWAS Catalog”. Page 6

5. Results from the MAGMA analysis described revealed in the female-specific analysis significant enrichment of gene-sets including “Hallmark gene-sets for Oestrogen response”, with “Early Oestrogen response” and “Late Oestrogen response” both being significant”. Taking this a bit further it is known that atherosclerosis progresses rapidly in women after menopause. There should be considerable number of women who have entered menopause in this sample aged between 40 and 60 years. There was no direct information on menopause collected in this study according to the study data dictionary. However, anticipating that the menopause is occurring at a very similar age in African women as in women in the US (including African American women) it would be possible to examine the effect of those SNPs on mean max cIMT stratifying on average age of menopause.

We agree with the reviewer that there is evidence that atherosclerosis progresses more rapidly in women after menopause. As pointed out, these studies have primarily been conducted in European populations and this correlation is yet to be robustly investigated in Africans. Our data has not focussed on age at menopause, although we did collect data on pre-, peri- and post-menopause in the cohort at baseline. We therefore ran an analysis of the female-specific significant variants (rs115473055, rs150840489) in two strata (pre-menopausal and post-menopausal women) and found significant differences in allele effect within post-menopausal women. A paragraph has been added in the results section: “In order to further investigate the effect of menopause in our sample, we tested the impact of menopause on female-specific significant variants (rs115473055, rs150840489) in two strata (pre-menopausal and post-menopausal women). We found that for rs115473055, the carriers of the T allele displayed higher cIMT values than those with the C allele when participants were post-menopausal. There was no difference in pre-menopausal women (Figure 6).” Page 7

6. minor comment is that all supplemental Tables need descriptive heading and directive notes.

This is important and we have added descriptive titles and brief notes.

Reviewer #3 (Remarks to the Author):

Boua et al. report the largest GWAS to date of carotid intima-media thickness (cIMT) in populations of African origins through the Africa Wits-INDEPTH Partnership for Genomic Studies cohort (AWI-Gen) which includes >12k participants from 4 sub-Saharan African countries. The current report includes a subset of nearly 8,000 participants with both genotype data using the H3Africa genotyping array followed by imputation onto the African Genome Resources panel (leading to ~14 million SNPs imputed after QC) and Mean Max cIMT measures (average of the maximum cIMT from the left and right). The investigators highlight the identification of 2 new genome-wide significant loci in the main analyses and 4 more ‘sex-specific’ loci.

Major strengths of the study include the populations under study given how overwhelmingly under-represented African populations have been in genomic studies to date, the availability of a high-density of genotyped markers to match the genetic diversity of the population (as well as to maximize the quality of the imputation when combined with the African Genome Resources panels), the sample size in the context of the effort needed to conduct the study, the use of BOLT LMM to perform analyses and guard against population stratification, and the compelling biology of the mapped genes identified by the GWAS.

That being said, there is room for improvement of the report as suggested below:

1) Title/abstract/intro: The link between cIMT measures alone and the biology of atherosclerosis is overstated. The potential contribution of one-time cIMT measure to risk of MI and stroke is modest with hazard ratios in the ~1.1 range (PMID: 22910757) and repeated cIMT measures have not been shown to be predictive of clinical events (PMID: 22541275, 29649236). These findings should be mentioned and the utility of cIMT in clinical practice should be briefly reviewed.

Thank you for this comment and the need to guard against overstating the implications for clinical atherosclerosis. The following text has been amended in the introduction:

“cIMT is a widely accepted surrogate marker for the risk of generalized atherosclerosis and is a measurement used in large-scale research studies on genetic associations with future cardiovascular events (Bartels, Franco, and Rundek 2012; Van Den Oord et al. 2013). Although a measurement of cIMT at one time point suggests a modest hazard ratio and low predictive ability for future clinical events (Den Ruijter et al. 2012; Lorenz et al. 2012), reference intervals of cIMT within populations are important and have been used for diagnostic purposes to predict risk, using the Mannheim Consensus guidelines (Touboul et al. 2013). Page 3

2) Intro: “To date, 76 SNPs have been found to be robustly associated with cIMT” – it would be helpful to summarize these findings better up front for the reader. How many loci do these SNPs represent? How many of these are also established loci for atherosclerosis (e.g. for CAD or ischemic stroke or PAD). Which racial/ethnic groups were these discoveries made in?

The statement has been updated to include more details: “To date, 130 SNPs have been found to be robustly associated with cIMT (GWAS Catalog) (Buniello et al., 2019). The studies that are reported in the GWAS Catalog are primarily from European-ancestry populations, with small numbers of Hispanic, African-American and Chinese participants. There are no studies on sub-Saharan African populations resident in Africa.” Page 3

3) Intro/rest of manuscript: I do not find the argument for sex-stratified analyses in the intro compelling. All but one of the references (Rawlik et al. 2016, Franconi et al. 2017, and Dong et al. 2015, and Lin et al. 2015) are low impact reports that do not provide convincing

evidence that the biology of cIMT is different among males vs. females. The most interesting reference is the Rawlik paper but that paper does not include cIMT as one of the phenotypes. Furthermore, the genetic correlations between males and females of a majority of the traits included in that study were still very high (>0.9) even if they were found to be statistically different from 1. This suggests differences between the sexes are quite subtle. The arguments for a sex-stratified analyses thus need to be tempered.

This point is somewhat controversial and the other reviewers expressed interest in this analysis. In addition, sex-stratified analysis of individuals from the UKBB have been reported where sex-specific loci for cIMT are described. We have added to the text to improve the argument for sex-stratified analysis: “A recent GWA study of cIMT reported sex-specific loci from analyses of women and men from the UKBB data (Strawbridge *et al.*, 2020).” Page 3

4) Methods: The authors refer to a design/methods paper for the cIMT protocol used (Ali *et al.*). It would be helpful if the authors could mention (or reference) the presence of expected cross-sectional associations between atherosclerosis risk factors (e.g. age, sex, lipid levels, blood pressure) and the cIMT measures in the cohorts. This would provide some reassurance of the data quality of the cIMT is high.

We have done extensive QC on the data and have published several papers on the associations with other classical risk factors (last paragraph of the introduction on page 4), and these have been included and an additional 2021 reference added to provide confidence in the quality of the cIMT data.

5) Methods/results: In relation to #3, the risk of false positive genetic associations increases as the sample size decreases. The investigators mention power calculations with QUANTO for gene-environment, but they do not actually present these calculations. Post-hoc power calculations for the main effect positive associations observed in the combined and the sex-specific analyses need to be presented given the likely low variance explained by each variant. I calculated power for identifying a variant explaining 0.5% of the variance of a quantitative trait using the Purcell’s genetic power calculator (<http://zzz.bwh.harvard.edu/gpc/>) and found very low power to detect such associations with the sample sizes available here. Low power increases the risk of false positives (see Wacholder *et al.* PMID: 15026468). Thus, power calculations need to be presented so the reader can assess probability of a false positive given the proportion of variance of cIMT explained by each SNP.

Thank you for this important comment. Power calculation was performed using Quanto for the combined sample analysis for a range of potential effect sizes (beta values for SNP effect), based on previous findings from the GWAS Catalog. We have added the following to the text for more clarity: “A model that assesses the cIMT in independent individuals with an additive genetic inheritance and an allele frequency of 0.04 will be > 93% powered ($\alpha = 5 \times 10^{-8}$) to detect a β_G (genetic effect) of 0.0147 mm. Likewise, an allele frequency of 0.48 will have 98% power to detect even a very small genetic effect ($\beta = 0.0067$ mm).” To illustrate these power calculations, we have added a supplementary figure, S Figure 4).

Page 13

6) Results: Arguing for the presence of genetic effects in one sex that are in the opposite direction in the other sex (e.g CROCC) for cIMT is challenging given no risk factors or genetic findings to date exist that support such findings for any manifestation of atherosclerosis. The possibility of these findings being false positives should be included in the presentation of these results.

Thank you for this comment. We added the following: “Interestingly, our study identified 89 SNPs with suggestive association ($P < 1E-04$) which had opposite effects on cIMT between men and women (led by signals in the CROCC locus). In contrast, the sex-stratified analysis with participants from the UKBB reported a single discordant sex effect and the remainder of the effect directions were concordant, it is however possible that sex differences are exacerbated in African populations and is therefore noteworthy. Although statistically significant (Supplementary Figure 2 of sex-difference test Manhattan plot) these findings need to be explored further.”

7) Methods: Have the authors considered a different p-value for genome wide significance in this African population given the overall higher number of variants genotyped/imputed combined with the lower LD? See for example arguments/data in PMID 26733288/27305981. Several of the findings are just barely genome wide significant using the traditional cut of 5×10^{-8} .

The reviewer is correct that a different cut-off threshold has been discussed in light of lower LD in Africans and the number of tested variants. We have included a paragraph in the strengths and limitations section to discuss this: “Although the threshold for genome-wide significance that would be considered appropriate for GWAS conducted in African populations is under debate (Gurdasani et al. 2019), in absence of a clear guideline we have employed the traditional genome wide significance threshold of $5E-08$.” Page 11

8) Results: The first part of the Results section tends to regurgitate findings in the tables by listing all data already in Tables 1-4. First, I suggest that this is minimized. Second, I believe it would be easier for the reader if only the significant findings are presented in one table and all the suggestive findings are sent to the supplement. The sex-difference findings are not that compelling (Table 4) given low power to detect differences between the groups.

We agree with the reviewer that we should focus on the significant finding in the body of the paper (Table 1) and relegate some of the suggestive findings to the supplement. We have adjusted the manuscript accordingly. We do, however, feel that the sex differences merit some prominence and hope that this will stimulate others to do further exploration.

9) Results/Discussion: The term “replication” is used too liberally throughout the manuscript including in the first paragraph of the discussion section when comparing results of this cIMT GWAS to those of “related” phenotypes in the GWAS catalog. Replication should only be used for findings from other studies using the exact same phenotype of cIMT. All other findings are simply evidence of pleiotropy which might be vertical or might be horizontal.

This comment is in line with comments from Reviewers 1 and 2 and we have taken this on board. The replication criteria have been reviewed for clarity and revised throughout the manuscript to avoid confusion.

10) The authors should consider a genetic correlation study between their findings and published findings in Europeans for cIMT (and related phenotypes) using such tools as POPCORN (PMID: 27321947). Such findings would be much more convincing of “replication” (or more specifically genetic overlap of biology). Similar genetic correlation analyses could be considered for males vs females within this study. Such analyses would have more power than single SNP analyses.

We thank the reviewer for this suggestion. We have amended the replication analysis based on the current and other reviewers suggestions.

11) The Discussion is too long with a large section of it repeating the information provided in Figure 4 with even more detail for each locus in the extended discussion. It would be better if the authors could summarize the key themes of function implicated by the genes so as to summarize/complement the info provided in Figure 4/supplement.

Thank you for pointing this out, we have revised the discussion to minimise repetition and to emphasise the key outcomes.

12) The “collider bias” argument is unclear to me. What exactly is the collider and what evidence is there that the collider leads to distorted associations in males vs females when one adjusts for sex?

We thank the reviewer for drawing this to our attention. The phrase was out of scope and has been deleted from the manuscript.

Reviewers' Comments:

Reviewer #1:

Remarks to the Author:

The revised paper is much improved and most of my comments have been addressed. A few more remaining issues are:

1. There is still too much detailed text on several suggestive associations. Lines 143-154 should be removed or replaced with a simple statement such as "Loci with suggestive associations ($p < 1E-06$) are presented in suppl table 3, suppl table 4, suppl figure 2).
2. Line 156-168: The qualitative evidence on sex-dimorphism is not convincing, and may be misleading given that the betas are closer to 0 even when they are opposite. I recommend the authors to remove this section altogether (line 156-168, methods, discussion).
3. Line 210-223: There is detailed text on look-up of cardiovascular trait loci based on SUGGESTIVE male- and female-specific signals from this study. This is not informing enough to be included in the paper because the signals are not significant and the look-up was made from a sex-specific GWAS in the present study to published sex-combined GWAS of cardiovascular traits. I suggest that line 210-223 be removed.

Reviewer #2:

Remarks to the Author:

my comments have been addressed and I have no additional comments

Reviewer #3:

Remarks to the Author:

The authors have been partially responsive to my comments/suggestions . I still would like a few of my concerns addressed more completely. Below is the subset of my original concerns that were not addressed fully

- 1) Title/abstract/intro: The link between cIMT measures alone and the biology of atherosclerosis is overstated. The potential contribution of one-time cIMT measure to risk of MI and stroke is modest with hazard ratios in the ~ 1.1 range (PMID: 22910757) and repeated cIMT measures have not been shown to be predictive of clinical events (PMID: 22541275, 29649236). These findings should be mentioned and the utility of cIMT in clinical practice should be briefly reviewed.

Thank you for this comment and the need to guard against overstating the implications for clinical atherosclerosis. The following text has been amended in the introduction:

"cIMT is a widely accepted surrogate marker for the risk of generalized atherosclerosis and is a measurement used in large-scale research studies on genetic associations with future cardiovascular events (Bartels, Franco, and Rundek 2012; Van Den Oord et al. 2013). Although a measurement of cIMT at one time point suggests a modest hazard ratio and low predictive ability for future clinical events (Den Ruijter et al. 2012; Lorenz et al. 2012), reference intervals of cIMT within populations are important and have been used for diagnostic purposes to predict risk, using the Mannheim Consensus guidelines (Touboul et al. 2013). Page 3

RESIDUAL CONCERNS: The new language is certainly an improvement, but it continues to suggest that there is a some role in clinical practice to measuring cIMT for risk prediction in CVD. However, there is no evidence to support the use of cIMT in anything other than research. I referred to PMID 22541275, 29649236 once again as 2 examples which were not incorporated into this response.

Mannheim Consensus guidelines are outdated and have very little info about using cIMT in clinical practice and that info is also outdated. I recommend removing the statement "reference intervals of cIMT within populations are important and have been used for diagnostic purposes to predict risk, using the Mannheim Consensus guidelines". I do not mind the statement that the cIMT measure has some relevance to understanding the biology of atherosclerosis (because the GWAS data from prior studies support this) but this measure has no proven role in "diagnosis" or "prognosis-risk prediction" in clinical practice incremental to standard risk factor profiling with clinical risk scores.

2) Intro: "To date, 76 SNPs have been found to be robustly associated with cIMT" – it would be helpful to summarize these findings better up front for the reader. How many loci do these SNPs represent? How many of these are also established loci for atherosclerosis (e.g. for CAD or ischemic stroke or PAD). Which racial/ethnic groups were these discoveries made in?

The statement has been updated to include more details: "To date, 130 SNPs have been found to be robustly associated with cIMT (GWAS Catalog) (Buniello et al., 2019). The studies that are reported in the GWAS Catalog are primarily from European-ancestry populations, with small numbers of Hispanic, African-American and Chinese participants. There are no studies on sub-Saharan African populations resident in Africa." Page 3

RESIDUAL CONCERNS: The authors have only updated the number of SNPs from 76 to 130 perhaps due to recent new entries in the GWAS Catalog. The number of SNPs still does not give the reader a clearer picture of the number of independent loci that these 130 SNPs represent. Is it 10 loci with 13 SNPs each or 65 loci with 2 SNPs in each locus etc.? The authors should clarify. They could run these 130 SNPs through programs that calculate LD between them to determine the number of loci after appropriate pruning. Alternatively, they could manually curate/ synthesize the literature (every SNP in the catalog is attached to a Pubmed ID). Also, each publication and the GWAS catalog informs on whether race/ethnic group the discovery was made in. This info should be included.

Lastly, please clarify what "robustly associated with cIMT" means. Is this at genome wide significance or at a lower level of significance given the GWAS catalogue lists SNP associations down to 10⁻⁵ p values?

3) Intro/rest of manuscript: I do not find the argument for sex-stratified analyses in the intro compelling. All but one of the references (Rawlik et al. 2016, Franconi et al. 2017, and Dong et al. 2015, and Lin et al. 2015) are low impact reports that do not provide convincing evidence that the biology of cIMT is different among males vs. females. The most interesting reference is the Rawlik paper but that paper does not include cIMT as one of the phenotypes. Furthermore, the genetic correlations between males and females of a majority of the traits included in that study were still very high (>0.9) even if they were found to be statistically different from 1. This suggests differences between the sexes are quite subtle. The arguments for a sex-stratified analyses thus need to be tempered.

This point is somewhat controversial and the other reviewers expressed interest in this analysis. In addition, sex-stratified analysis of individuals from the UKBB have been reported where sex-specific loci for cIMT are described. We have added to the text to improve the argument for sex-stratified analysis: "A recent GWA study of cIMT reported sex-specific loci from analyses of women and men from the UKBB data (Strawbridge et al., 2020)." Page 3

RESIDUAL CONCERNS: I do not object to maintaining the sex-specific analyses in the paper, but I think the authors still have not placed the hypothesis into better perspective in the introduction and in the discussion. In the background, there is not much evidence of a sex specific biology of atherosclerosis. Females have less of it compared to males until about the age of 60 but this does not mean that the biology is different between the sexes. A large majority of attempts to identify sex-

specific mechanisms has shown that all major risk factors are identical and effect sizes are comparable, and certainly not in the opposite direction. A large fraction of the difference in incidence/prevalence of atherosclerosis between males and females can be explained by higher prevalence of the exact same risk factors in males compared to females (see for example INTERHEART study publications). Thus, in my opinion, identifying sex-specific genetic mechanisms in a relatively small cohort for a phenotype that is arguably a proxy for atherosclerosis sets up the possibility of false positives if the study is not well powered. Thus, the current work is at best hypothesis generating.

The UK biobank study is not cited appropriately in my opinion. First, multiple sex-specific loci are NOT reported in that study. That study actually finds little that is sex specific – the authors of that statement make a highly qualitative statement that “suggest that the genetic variants associated with IMTmean in men and women are distinctly different but then the interaction results actually highlight that a majority of findings are directionally consistent and/or consistent with the primary combined (male+female) analysis with the possible exception of just 1 female specific locus. These paucity of findings in a study that is almost 3 times larger than this one. Unfortunately, that study didn’t report genetic correlations between male and female specific loci but correlation with . I suggest the authors check the global genetic correlation in their own data between males and females. I am quite confident it will be high and this will also add perspective to the other findings.

5) Methods/results: In relation to #3, the risk of false positive genetic associations increases as the sample size decreases. The investigators mention power calculations with QUANTO for gene-environment, but they do not actually present these calculations. Post-hoc power calculations for the main effect positive associations observed in the combined and the sex-specific analyses need to be presented given the likely low variance explained by each variant. I calculated power for identifying a variant explaining 0.5% of the variance of a quantitative trait using the Purcell’s genetic power calculator (<http://zzz.bwh.harvard.edu/gpc/>) and found very low power to detect such associations with the sample sizes available here. Low power increases the risk of false positives (see Wacholder et al. PMID: 15026468). Thus, power calculations need to be presented so the reader can assess probability of a false positive given the proportion of variance of cIMT explained by each SNP.

Thank you for this important comment. Power calculation was performed using Quanto for the combined sample analysis for a range of potential effect sizes (beta values for SNP effect), based on previous findings from the GWAS Catalog. We have added the following to the text for more clarity: “A model that assesses the cIMT in independent individuals with an additive genetic inheritance and an allele frequency of 0.04 will be > 93% powered ($\alpha = 5 \times 10^{-8}$) to detect a β_G (genetic effect) of 0.0147 mm. Likewise, an allele frequency of 0.48 will have 98% power to detect even a very small genetic effect ($\beta = 0.0067$ mm).” To illustrate these power calculations, we have added a supplementary figure, S Figure 4). Page 13

RESIDUAL CONCERNS: The power figure is a step in the right direction but I am still left without a good sense of the post-hoc power to detect the associations reported in Table 1. The statement gives the impression of good power but the combination of allele frequencies and genetic effects in the statement are not in line with those reported in Table 1. The authors should calculate the exact power for their findings in Table 1, not for findings in the GWAS catalogue -- i.e. the power to detect a genetic effect of 0.043 with a MAF of 0.013 with a sample size of ~7900 (rs552690895) is XX% and include it in a final column labelled “Post-hoc power”. If this power is <50% then the probability of a false positive increases substantially and this should be clearly stated in the discussion along with the need for extensive validation of these findings.

6) Results: Arguing for the presence of genetic effects in one sex that are in the opposite direction in the other sex (e.g CROCC) for cIMT is challenging given no risk factors or genetic findings to date exist that support such findings for any manifestation of atherosclerosis. The possibility of these findings being false positives should be included in the presentation of these results.

Thank you for this comment. We added the following: "Interestingly, our study identified 89 SNPs with suggestive association ($P < 1E-04$) which had opposite effects on cIMT between men and women (led by signals in the CROCC locus). In contrast, the sex-stratified analysis with participants from the UKBB reported a single discordant sex effect and the remainder of the effect directions were concordant, it is however possible that sex differences are exacerbated in African populations and is therefore noteworthy. Although statistically significant (Supplementary Figure 2 of sex-difference test Manhattan plot) these findings need to be explored further."

RESIDUAL CONCERNS: In the absence of power calculations to detect sex interactions observed, it is my impression that the interaction analyses are also substantially underpowered (even more than the main effects). The authors continue to avoid discussing this possibility openly/directly. I think the statement "need to be explored further" is too vague. A better statement would be "Although statistically significant, these findings need to be replicated in additional studies because our power to detect the associations observed was low, substantially increasing the likelihood that these associations we observed are false positives."

REVIEWER COMMENTS

We thank the reviewers for their helpful comments. Our responses are shown below in blue text. The changes to the manuscript have been highlighted in yellow.

Reviewer #1 (Remarks to the Author):

The revised paper is much improved and most of my comments have been addressed. A few more remaining issues are:

1. There is still too much detailed text on several suggestive associations. Lines 143-154 should be removed or replaced with a simple statement such as "Loci with suggestive associations ($p < 1E-06$) are presented in suppl table 3, suppl table 4, suppl figure 2).

Response 1: We have reduced the details and referred the readers to the supplemental materials. We have amended the text as following: "Loci with suggestive sex-specific associations are shown in Supplementary Table 3". 2nd paragraph, Page 5

2. Line 156-168: The qualitative evidence on sex-dimorphism is not convincing, and may be misleading given that the betas are closer to 0 even when they are opposite. I recommend the authors to remove this section altogether (line 156-168, methods, discussion).

Response 2: We understand that this section needs more clarification and should be framed more as an hypothesis generating statement, since our sample size is relatively small. However, we consider that this is an important finding. Throughout the paper, the beta values are expressed in mm and are in the range of a variance of 2% to 8% in the sex-dimorphism section, which is not negligible. Additionally, this section provides a second report of evidence of sex differences in cIMT, to support the findings first reported in UKBB, and for that reason we prefer to keep this section in the paper. To clarify this further, we have added in the introduction section: "Despite the success of GWAS efforts, men and women have typically been analyzed together in sex-combined analyses, with sex used as a covariate in the model to account for marginal differences on traits between them. Sex-combined analyses assume homogeneity of the allelic effects in men and women, and therefore are sub-optimal in the presence of heterogeneity in genetic effects by sex, i.e., sex-dimorphic effects." 4th paragraph, Page 3.

3. Line 210-223: There is detailed text on look-up of cardiovascular trait loci based on SUGGESTIVE male- and female-specific signals from this study. This is not informing enough to be included in the paper because the signals are not significant and the look-up was made from a sex-specific GWAS in the present study to published sex-combined GWAS of cardiovascular traits. I suggest that line 210-223 be removed.

Response 3: We thank the reviewer for this suggestion. We have removed the section from the manuscript.

Reviewer #2 (Remarks to the Author):

my comments have been addressed and I have no additional comments

Reviewer #3 (Remarks to the Author):

The authors have been partially responsive to my comments/suggestions . I still would like a few of my concerns addressed more completely. Below is the subset of my original concerns that were not addressed fully

1) Title/abstract/intro: The link between cIMT measures alone and the biology of atherosclerosis is overstated. The potential contribution of one-time cIMT measure to risk of MI and stroke is modest with hazard ratios in the ~1.1 range (PMID: 22910757) and repeated cIMT measures have not been shown to be predictive of clinical events (PMID: 22541275, 29649236). These findings should be mentioned and the utility of cIMT in clinical practice should be briefly reviewed.

Thank you for this comment and the need to guard against overstating the implications for clinical

atherosclerosis. The following text has been amended in the introduction:

"cIMT is a widely accepted surrogate marker for the risk of generalized atherosclerosis and is a measurement used in large-scale research studies on genetic associations with future cardiovascular events (Bartels, Franco, and Rundek 2012; Van Den Oord et al. 2013). Although a measurement of cIMT at one time point suggests a modest hazard ratio and low predictive ability for future clinical events (Den Ruijter et al. 2012; Lorenz et al. 2012), reference intervals of cIMT within populations are important and have been used for diagnostic purposes to predict risk, using the Mannheim Consensus guidelines (Touboul et al. 2013). Page 3

RESIDUAL CONCERNS: The new language is certainly an improvement, but it continues to suggest that there is a some role in clinical practice to measuring cIMT for risk prediction in CVD. However, there is no evidence to support the use of cIMT in anything other than research. I referred to PMID 22541275, 29649236 once again as 2 examples which were not incorporated into this response. Mannheim Consensus guidelines are outdated and have very little info about using cIMT in clinical practice and that info is also outdated. I recommend removing the statement "reference intervals of cIMT within populations are important and have been used for diagnostic purposes to predict risk, using the Mannheim Consensus guidelines". I do not mind the statement that the cIMT measure has some relevance to understanding the biology of atherosclerosis (because the GWAS data from prior studies support this) but this measure has no proven role in "diagnosis" or "prognosis-risk prediction" in clinical practice incremental to standard risk factor profiling with clinical risk scores.

Response 1: Thank you for this comment and the need to guard against overstating the implications for clinical atherosclerosis, we have removed the text related to the utility of cIMT in clinical practice. 2nd paragraph, Page 3

2) Intro: "To date, 76 SNPs have been found to be robustly associated with cIMT" – it would be helpful to summarize these findings better up front for the reader. How many loci do these SNPs represent? How many of these are also established loci for atherosclerosis (e.g. for CAD or ischemic stroke or PAD). Which racial/ethnic groups were these discoveries made in?

The statement has been updated to include more details: "To date, 130 SNPs have been found to be robustly associated with cIMT (GWAS Catalog) (Buniello et al., 2019). The studies that are reported in the GWAS Catalog are primarily from European-ancestry populations, with small numbers of Hispanic, African-American and Chinese participants. There are no studies on sub-Saharan African populations resident in Africa." Page 3

RESIDUAL CONCERNS: The authors have only updated the number of SNPs from 76 to 130 perhaps due to recent new entries in the GWAS Catalog. The number of SNPs still does not give the reader a clearer picture of the number of independent loci that these 130 SNPs represent. Is it 10 loci with 13 SNPs each or 65 loci with 2 SNPs in each locus etc.? The authors should clarify. They could run these 130 SNPs through programs that calculate LD between them to determine the number of loci after appropriate pruning. Alternatively, they could manually curate/ synthesize the literature (every SNP in the catalog is attached to a Pubmed ID). Also, each publication and the GWAS catalog informs on whether race/ethnic group the discovery was made in. This info should be included.

Lastly, please clarify what "robustly associated with cIMT" means. Is this at genome wide significance or at a lower level of significance given the GWAS catalogue lists SNP associations down to 10⁻⁵ p values?

Response 2: We thank the reviewer for their comments, the section has been amended : "To date, 136 SNPs from 98 independent loci have been found to be associated with cIMT (GWAS Catalog) regardless of ancestry (24). The loci were reported for cIMT, in the presence or absence of gene-environment interactions (Gene x HIV, Gene x Smoking, Gene x Sex, Gene x Rheumatoid arthritis). The studies that are reported in the GWAS Catalog are primarily from European-ancestry populations, with small numbers of Hispanic, African-American and Chinese participants. There was only one study on sub-Saharan African populations resident in Africa (25).."

3) Intro/rest of manuscript: I do not find the argument for sex-stratified analyses in the intro

compelling. All but one of the references (Rawlik et al. 2016, Franconi et al. 2017, and Dong et al. 2015, and Lin et al. 2015) are low impact reports that do not provide convincing evidence that the biology of cIMT is different among males vs. females. The most interesting reference is the Rawlik paper but that paper does not include cIMT as one of the phenotypes. Furthermore, the genetic correlations between males and females of a majority of the traits included in that study were still very high (>0.9) even if they were found to be statistically different from 1. This suggests differences between the sexes are quite subtle. The arguments for a sex-stratified analyses thus need to be tempered.

This point is somewhat controversial and the other reviewers expressed interest in this analysis. In addition, sex-stratified analysis of individuals from the UKBB have been reported where sex-specific loci for cIMT are described. We have added to the text to improve the argument for sex-stratified analysis: "A recent GWA study of cIMT reported sex-specific loci from analyses of women and men from the UKBB data (Strawbridge et al., 2020)." Page 3

RESIDUAL CONCERNS: I do not object to maintaining the sex-specific analyses in the paper, but I think the authors still have not placed the hypothesis into better perspective in the introduction and in the discussion. In the background, there is not much evidence of a sex specific biology of atherosclerosis. Females have less of it compared to males until about the age of 60 but this does not mean that the biology is different between the sexes. A large majority of attempts to identify sex-specific mechanisms has shown that all major risk factors are identical and effect sizes are comparable, and certainly not in the opposite direction. A large fraction of the difference in incidence/prevalence of atherosclerosis between males and females can be explained by higher prevalence of the exact same risk factors in males compared to females (see for example INTERHEART study publications). Thus, in my opinion, identifying sex-specific genetic mechanisms in a relatively small cohort for a phenotype that is arguably a proxy for atherosclerosis sets up the possibility of false positives if the study is not well powered. Thus, the current work is at best hypothesis generating.

The UK biobank study is not cited appropriately in my opinion. First, multiple sex-specific loci are NOT reported in that study. That study actually finds little that is sex specific – the authors of that statement make a highly qualitative statement that "suggest that the genetic variants associated with IMTmean in men and women are distinctly different but then the interaction results actually highlight that a majority of findings are directionally consistent and/or consistent with the primary combined (male+female) analysis with the possible exception of just 1 female specific locus. These paucity of findings in a study that is almost 3 times larger than this one. Unfortunately, that study didn't report genetic correlations between male and female specific loci but correlation with . I suggest the authors check the global genetic correlation in their own data between males and females. I am quite confident it will be high and this will also add perspective to the other findings.

Response 3: We thank the reviewer for further comments on the sex-specific analysis and highlighting the need to further temper the discussion. Even though the evidence can be criticised in light of our "modest" sample size, compelling epidemiological evidence from the same cohort found that the sex-difference is not explained by the differences in prevalence of the risk factors that were measured (although there may be other unmeasured risk factors) [PMID: 31304842, PMID: 33563289]. Additionally, consistent direction in effect are the major type sex-differential effect in genetic studies and is in accordance of the findings from both our current study and the study from UKBB. Future studies will allow confirmation of our findings. Amendments have been made in the introduction to place the hypothesis into better perspective, and the discussion has been tempered to considering the post-hoc power and to suggest future validation studies.

5) Methods/results: In relation to #3, the risk of false positive genetic associations increases as the sample size decreases. The investigators mention power calculations with QUANTO for gene-environment, but they do not actually present these calculations. Post-hoc power calculations for the main effect positive associations observed in the combined and the sex-specific analyses need to be presented given the likely low variance explained by each variant. I calculated power for identifying a variant explaining 0.5% of the variance of a quantitative trait using the Purcell's genetic power calculator (<http://zzz.bwh.harvard.edu/gpc/>) and found very low power to detect such associations with the sample sizes available here. Low power increases the risk of false positives (see Wacholder et al. PMID: 15026468). Thus, power calculations need to be presented so the reader can assess probability of a false positive given the proportion of variance of cIMT

explained by each SNP.

Thank you for this important comment. Power calculation was performed using Quanto for the combined sample analysis for a range of potential effect sizes (beta values for SNP effect), based on previous findings from the GWAS Catalog. We have added the following to the text for more clarity: "A model that assesses the cIMT in independent individuals with an additive genetic inheritance and an allele frequency of 0.04 will be > 93% powered ($\alpha = 5 \times 10^{-8}$) to detect a βG (genetic effect) of 0.0147 mm. Likewise, an allele frequency of 0.48 will have 98% power to detect even a very small genetic effect ($\beta = 0.0067$ mm)." To illustrate these power calculations, we have added a supplementary figure, S Figure 4). Page 13

RESIDUAL CONCERNS: The power figure is a step in the right direction but I am still left without a good sense of the post-hoc power to detect the associations reported in Table 1. The statement gives the impression of good power but the combination of allele frequencies and genetic effects in the statement are not in line with those reported in Table 1. The authors should calculate the exact power for their findings in Table 1, not for findings in the GWAS catalogue -- i.e. the power to detect a genetic effect of 0.043 with a MAF of 0.013 with a sample size of ~7900 (rs552690895) is XX% and include it in a final column labelled "Post-hoc power". If this power is <50% then the probability of a false positive increases substantially and this should be clearly stated in the discussion along with the need for extensive validation of these findings.

Response 5: We thank the reviewer for their suggestion. We have performed post-hoc power calculations based on the GWAS significant results, using power add (<https://rpubs.com/maffleur/post-hoc-power>) and the results are below. We have added in the strength and limitations section: "Post-hoc power calculation showed that three of the six genome-wide significant variants were powered (>50%) for discovery, suggesting the need for validation."

X.	rsID	chr	pos	non.effect.allele	effect.allele	MAF	gwasP	Beta	SE	Nearest.Ge	N	power_5e8	power_5e2
Combined analysis	rs552690895	5	1,08E+08	A	G	0,013	2,5E-08	-0,043	0	FBXL17	7894	0,465686	0,999669
	#N/A rs6045318	20	1883451	G	A	0,024	4,7E-08	-0,031	0	SIRPA	7894	0,384624	0,999308
Female specific	rs115473055	1	1,11E+08	T	C	0,08	1E-08	-0,026	0	CYMP	3963	0,39398	0,999364
	#N/A rs150840489	15	71088277	A	G	0,037	2,4E-09	-0,051	0	RPL29P30	3963	0,576339	0,999885
Male specific	rs190770959	16	12158574	T	C	0,015	6,3E-09	-0,056	0	SNX29	3931	0,550326	0,999851
	#N/A rs147978408	16	12171475	C	T	0,016	6,6E-09	-0,055	0	SNX29	3931	0,511029	0,999783

6) Results: Arguing for the presence of genetic effects in one sex that are in the opposite direction in the other sex (e.g CROCC) for cIMT is challenging given no risk factors or genetic findings to date exist that support such findings for any manifestation of atherosclerosis. The possibility of these findings being false positives should be included in the presentation of these results.

Thank you for this comment. We added the following: "Interestingly, our study identified 89 SNPs with suggestive association ($P < 1 \times 10^{-4}$) which had opposite effects on cIMT between men and women (led by signals in the CROCC locus). In contrast, the sex-stratified analysis with participants from the UKBB reported a single discordant sex effect and the remainder of the effect directions were concordant, it is however possible that sex differences are exacerbated in African populations and is therefore noteworthy. Although statistically significant (Supplementary Figure 2 of sex-difference test Manhattan plot) these findings need to be explored further."

RESIDUAL CONCERNS: In the absence of power calculations to detect sex interactions observed, it is my impression that the interaction analyses are also substantially underpowered (even more than the main effects). The authors continue to avoid discussing this possibility openly/directly. I think the statement "need to be explored further" is too vague. A better statement would be "Although statistically significant, these findings need to be replicated in additional studies because our power to detect the associations observed was low, substantially increasing the likelihood that these associations we observed are false positives."

Response 6: We thank the reviewer for this suggestion. The statement has been amended to : "Although statistically significant (Supplementary Figure 2 of sex-difference test Manhattan plot), these findings need to be explored further through replication in

additional studies because our power to detect the associations observed was low, substantially increasing the likelihood that the associations we observed are false positives.”

Reviewers' Comments:

Reviewer #3:

Remarks to the Author:

The authors have been very responsive to my suggestions after their first revision with one minor exception. I would like them to include somewhere in the manuscript the post-hoc power calculations they have shown in their response. I think I only see a qualitative statement (which I comment on below). I suggested these power calculations be included in Table 1 but including them as a supplementary table is another option.

I think the author qualitative statement about power: "Post-hoc power calculation showed that three of the six genome-wide significant variants were powered (>50%) for discovery, suggesting the need for validation" is a little misleading. A well powered analysis is >90% power, not >50%. A better statement is "Post-hoc power calculation showed that the power to discover the six genome-wide significant variants was modest, suggesting the need for validation"

REVIEWER COMMENTS

We thank the reviewers for their helpful comments. Our responses are shown below in blue text. The changes to the manuscript have been highlighted in yellow.

Reviewer #3 (Remarks to the Author):

REVIEWERS' COMMENTS

Reviewer #3 (Remarks to the Author):

The authors have been very responsive to my suggestions after their first revision with one minor exception. I would like them to include somewhere in the manuscript the post-hoc power calculations they have shown in their response. I think I only see a qualitative statement (which I comment on below). I suggested these power calculations be included in Table 1 but including them as a supplementary table is another option.

I think the author qualitative statement about power: "Post-hoc power calculation showed that three of the six genome-wide significant variants were powered (>50%) for discovery, suggesting the need for validation" is a little misleading. A well powered analysis is >90% power, not >50%. A better statement is "Post-hoc power calculation showed that the power to discover the six genome-wide significant variants was modest, suggesting the need for validation"

Response to reviewer

We thank the reviewer for their suggestion. We have amended the statement to: "Post-hoc power calculation showed that the power to discover the six genome-wide significant variants was modest (Supplementary Table 8), suggesting the need for validation". Page 10